# AGAV-Rater: Adapting Large Multimodal Model for AI-Generated Audio-Visual Quality Assessment

Yuqin Cao [1]   Xiongkuo Min [1]   Yixuan Gao [1]   Wei Sun [2]   Guangtao Zhai [1]

## Abstract

Many video-to-audio (VTA) methods have been proposed for dubbing silent AI-generated videos. An efficient quality assessment method for AI-generated audio-visual content (AGAV) is crucial for ensuring audio-visual quality. Existing audio-visual quality assessment methods struggle with unique distortions in AGAVs, such as unrealistic and inconsistent elements. To address this, we introduce **AGAVQA-3k**, the first large-scale AGAV quality assessment dataset, comprising 3,382 AGAVs from 16 VTA methods. AGAVQA-3k includes two subsets: AGAVQA-MOS, which provides multi-dimensional scores for audio quality, content consistency, and overall quality, and AGAVQA-Pair, designed for optimal AGAV pair selection. We further propose **AGAV-Rater**, a LMM-based model that can score AGAVs, as well as audio and music generated from text, across multiple dimensions, and selects the best AGAV generated by VTA methods to present to the user. AGAV-Rater achieves state-of-the-art performance on AGAVQA-3k, Text-to-Audio, and Text-to-Music datasets. Subjective tests also confirm that AGAV-Rater enhances VTA performance and user experience. The dataset and code are available at https://github.com/charlotte9524/AGAV-Rater.

This work was supported in part by the National Natural Science Foundation of China under Grant 62271312, Grant 62132006, Grant 62225112 and Grant 62301316; in part by STCSM under Grant 22DZ2229005; and in part by the Oceanic Interdisciplinary Program of Shanghai Jiao Tong University (project number SL2020ZD102). [1]Institute of Image Communication and Network Engineering, Shanghai Key Laboratory of Digital Media Processing and Transmissions, Shanghai Jiao Tong University, Shanghai [2]School of Communication & Electronic Engineering, East China Normal University, Shanghai. Correspondence to: Guangtao Zhai <zhaiguangtao@sjtu.edu.cn>, Xiongkuo Min <minxiongkuo@sjtu.edu.cn>.

*Proceedings of the 42nd International Conference on Machine Learning*, Vancouver, Canada. PMLR 267, 2025. Copyright 2025 by the author(s).

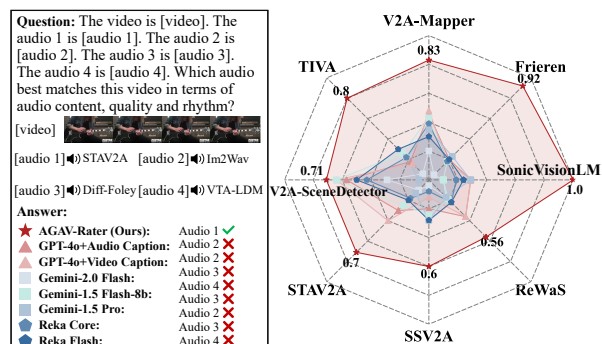

*Figure 1.* Comparison of answer accuracy between **AGAV-Rater** and proprietary LMMs on AGAV-Pair subset. We construct question-answer pairs from demonstration AGAVs on VTA GitHub pages, prompting LMMs to identify the optimal AGAV pair. Each dimension in the radar chart represents the answer accuracy with the correct answer corresponding to the current VTA method.

## 1. Introduction

*The quality of AI-generated content needs to be controlled.* Many researchers focus on developing video-to-audio (VTA) methods to add sound to silent AI-generated videos. Some researchers (Wang et al., 2024f; Lu et al., 2024) combine large multimodal models (LMMs) with diffusion models, empowering them with the ability to generate audio from videos. On the commercial side, companies like ElevenLabs have launched efficient VTA models to dub videos. Although VTA methods can significantly improve post-production efficiency and enhance the audio-visual (A/V) experience of AI-generated content (AIGC), they occasionally encounter issues such as poor A/V alignment or low perceptual audio quality when generating audio. Therefore, there is a need for an automated model to evaluate AI-generated audio-visual content (AGAV), select the most user-preferred results to present to users, and provide feedback for improving the generated content.

Traditional audio-visual quality assessment (AVQA) methods (Cao et al., 2023a;b; Min et al., 2020) focus on distortions caused during the capture and transmission stages, which makes it difficult to identify unique distortions in AIGC, such as inconsistent A/V content and unnatural audio. With the rise of LMMs, researchers (Wang et al., 2024b;c; Wu et al.) utilize the powerful content and language com-

prehension capabilities of LMMs to evaluate the quality of AIGC images and videos more accurately. However, most quality assessment research focuses on the visual capabilities of LMMs (Sun et al., 2024b;a; 2023), with little exploration of their A/V capabilities. This raises the question:

*Can LMMs be utilized to evaluate the quality of audio-visual content generated by VTA methods?*

In this paper, to develop and refine methods for evaluating AGAV quality, we construct the first AI-generated audio-visual quality assessment dataset, **AGAVQA-3k**, which contains $3,382$ AGAVs generated by 16 VTA methods. Fig. 2 illustrates the dataset construction pipeline. We hope that AGAV quality assessment methods can score AGAVs from multiple dimensions, while also assisting VTA models in selecting the optimal result from multiple generated outputs. To meet the above two needs, our AGAVQA-3k dataset is divided into two subsets. In the AGAVQA-MOS subset, we utilize 8 VTA methods to generate $3,088$ AGAV from 386 AIGC videos. Then we conduct a subjective experiment to collect $9,264$ Mean Opinion Scores (MOSs) across three dimensions, including audio perceptual quality, A/V content consistency, and overall A/V quality. In the AGAVQA-Pair subset, we collect 294 AGAVs from 8 VTA GitHub pages. The same video content forms a group, resulting in 75 question-answer pairs. We evaluate 7 closed-source LMMs on the AGAVQA-Pair subset to assess accuracy in selecting the optimal AGAVs, as shown in Fig. 1. The results indicate that LMMs still have significant room for improvement in evaluating AGAV quality. Moreover, existing closed-source LMMs have difficulty providing a numerical score for AGAV quality as humans do. Therefore, this paper primarily focuses on:

*How to adapt LMMs to score AGAV like humans?*

We propose the first LMM-based quality assessment method for AGAV, **AGAV-Rater**. AGAV-Rater is trained through two stages to perceive AGAV in a human-like manner and outputs numerical scores across three dimensions. Firstly, we create $50,952$ instruction-response pairs related to the perceived quality from 3 large-scale real-world audio-caption datasets, including audio-visual datasets VGGSound (Chen et al., 2020), audio captioning dataset AudioCaps (Kim et al., 2019), and music captioning dataset MusicCaps (Agostinelli et al., 2023). These instruction-response pairs do not require human annotations. Instead, the responses are automatically labeled using two text-defined rating levels (excellent and bad). These labels are then utilized to pretrain the LMM, enabling it to roughly assess whether the quality is good or bad. This approach significantly reduces the labor and costs associated with dataset construction and allows the model to better predict numerical scores in subsequent stages. Finally, we fine-tune the pre-trained LMM

on human-annotated multi-dimensional MOSs.

Our experimental results demonstrate that AGAV-Rater achieves state-of-the-art performance on three quality assessment datasets: AGAVQA-MOS, text-to-audio (TTA), and text-to-music (TTM) (Deshmukh et al., 2024). Since the video content and VTA methods in the AGAVQA-MOS and AGAVQA-Pair subsets do not overlap, we validate the generalization ability and robustness of AGAV-Rater on the unseen dataset AGAVQA-Pair, as shown in Fig. 1. We further conduct a subjective experiment and find that AGAV-Rater helps VTA methods select high-quality generated results to present to users, thereby enhancing user experience. Our core contributions are threefold:

- **A large-scale AGAV quality assessment dataset AGAVQA-3k**. It labels AGAVs' quality in two ways: multi-dimensional numerical scores and the optimal AGAV pair.
- **A novel LMM-based AGAV quality assessment model, AGAV-Rater**. It can predict multi-dimensional quality scores for AGAVs, TTA, and TTM, and assist VTA methods in selecting the optimal AGAV samples.
- **Enhance the user experience of AGAVs generated by VTA methods.** According to our experiment, $80\%$ of users recognize that using AGAV-Rater to select higher-quality AGAVs offers a better A/V experience, validating that AGAV-Rater can assist VTA methods in improving quality.

## 2. Related Works

### 2.1. Audio-Visual Quality Assessment Dataset

Early research focused on compression distortions during transmission, leading to the construction of several traditionally distorted AVQA datasets. The largest one is the LIVE-SJTU dataset proposed by Min *et al.* (Min et al., 2020), which contains 14 original high-quality A/V consequences and 336 degraded ones. With the development of streaming media, researchers (Cao et al., 2023b; Ying et al., 2022) found that user-uploaded videos have more complex and diverse distortions, thus constructing authentically distorted AVQA datasets. With the rise of generative models, AIGC exhibits unique distortions that do not occur in real-world scenarios. To explore the distortions and perceived quality of AGAVs, we establish the first AGAV quality assessment dataset, AGAVQA-3k. Compared to single-modal AIGC image (video) quality assessment datasets, AGAVQA-3k dataset tackles more complex multimodal challenges, such as A/V content inconsistency and synchronization issues.

### 2.2. Quality Assessment Methods

**Audio Quality Assessment.** Most audio quality assessment (AQA) methods focus on distortions in speech recordings

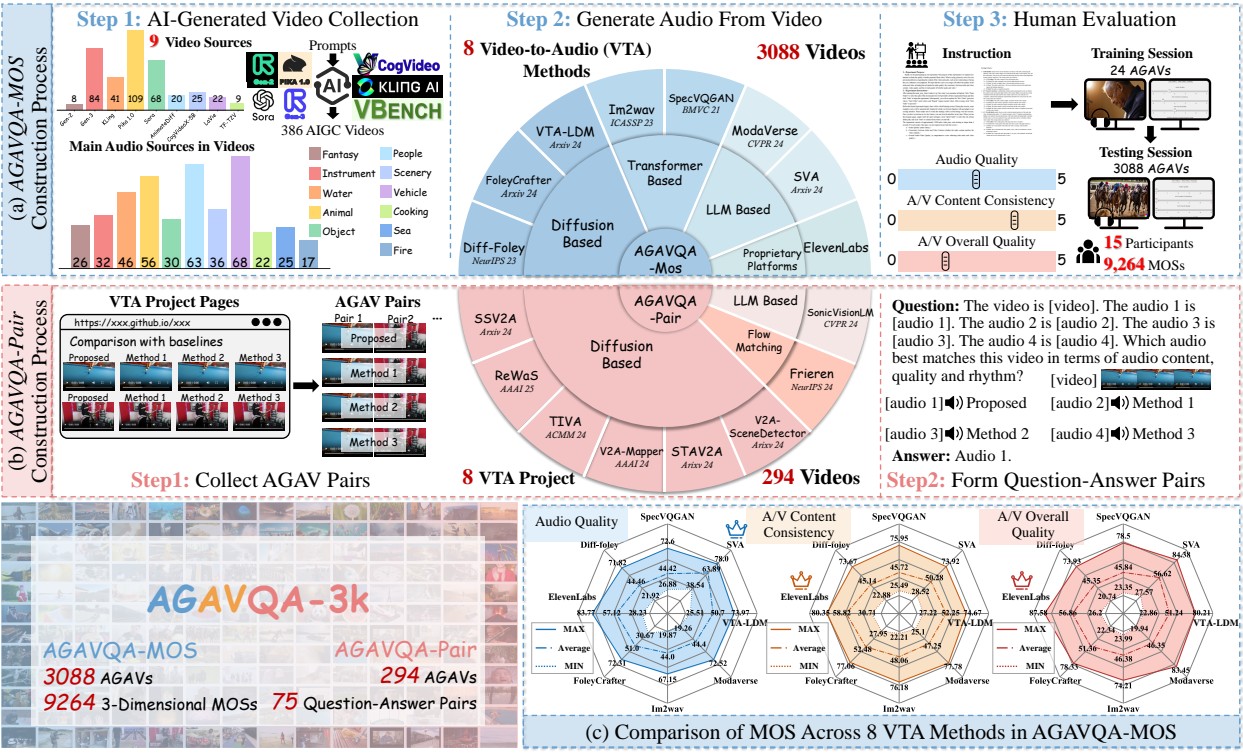

*Figure 2.* The construction process of the AGAVQA-3k dataset. The AGAVQA-3k dataset is divided into two subsets: (a) AGAVQA-MOS and (b) AGAVQA-Pair, which involve multi-dimensional score prediction and optimal AGAV pair selection tasks. (c) We present the maximum, minimum, and average subjective scores for the 8 VTA methods across the three dimensions.

from modern communication networks, such as noise and discontinuities. Early speech quality assessments (Rix et al., 2001; Beerends et al., 2013) used handcrafted metrics designed by speech experts to predict speech quality. However, these methods rely on comparing degraded speech with clean references, severely limiting their application in real-world scenarios. As a result, researchers proposed machine learning-based methods to predict speech quality using only degraded speech, eliminating the need for clean speech references during inference. Training a robust speech quality evaluator requires large-scale listening tests to collect speech and MOS for training. For example, NORESQA-MOS (Manocha & Kumar, 2022) was trained on 7,000 audio recordings, and NISQA (Mittag et al., 2021) was trained on 72,903 audio files. Soham *et al.* (Deshmukh et al., 2024) applied the original weight of audio-language models directly to predict TTA and TTM quality without additional training data, which led to limited performance. Despite the high cost of collecting training samples, most AQA methods have difficulty handling the unique distortions in AI-generated audio, which differ from real-world audio distortions. In this paper, we construct 50,952 instruction-response pairs without human annotations for pre-training the AGAV-Rater, thereby alleviating the burden of large-scale subjective experiments while allowing the AGAV-Rater to capture distortions in AIGC.

**Audio-Visual Quality Assessment.** Compared to AQA, AVQA is a more complex task as it requires handling the interaction between video and audio modalities. Early research (Min et al., 2020) utilized Support Vector Regression (SVR) to predict A/V quality scores by regressing handcrafted features extracted from video and audio. Although this method is effective for evaluating compressed A/V content, it performs poorly in real-world scenarios with mixed distortions. To address this issue, researchers (Cao et al., 2023b) have proposed deep learning-based approaches to predict A/V quality in real-world environments. However, these methods are not suitable for AGAVs, as they do not consider A/V content consistency and fail to address unnatural audio often present in AIGC scenarios. Our proposed AGAV-Rater not only predicts the quality of AGAVs but also evaluates the quality of TTAs and TTMs. We validate the effectiveness of AGAV-Rater on the AGAVQA-3k, TTA, and TTM datasets.

## 3. Dataset Construction

Our proposed AGAVQA-3k dataset consists of two subsets: the AGAVQA-MOS and AGAVQA-Pair, designed for multi-dimensional score prediction and optimal AGAV selection, respectively. In this section, we introduce the construction process of two subsets and analyze the subjective scores in the AGAVQA-MOS subset, as illustrated in Fig. 2.

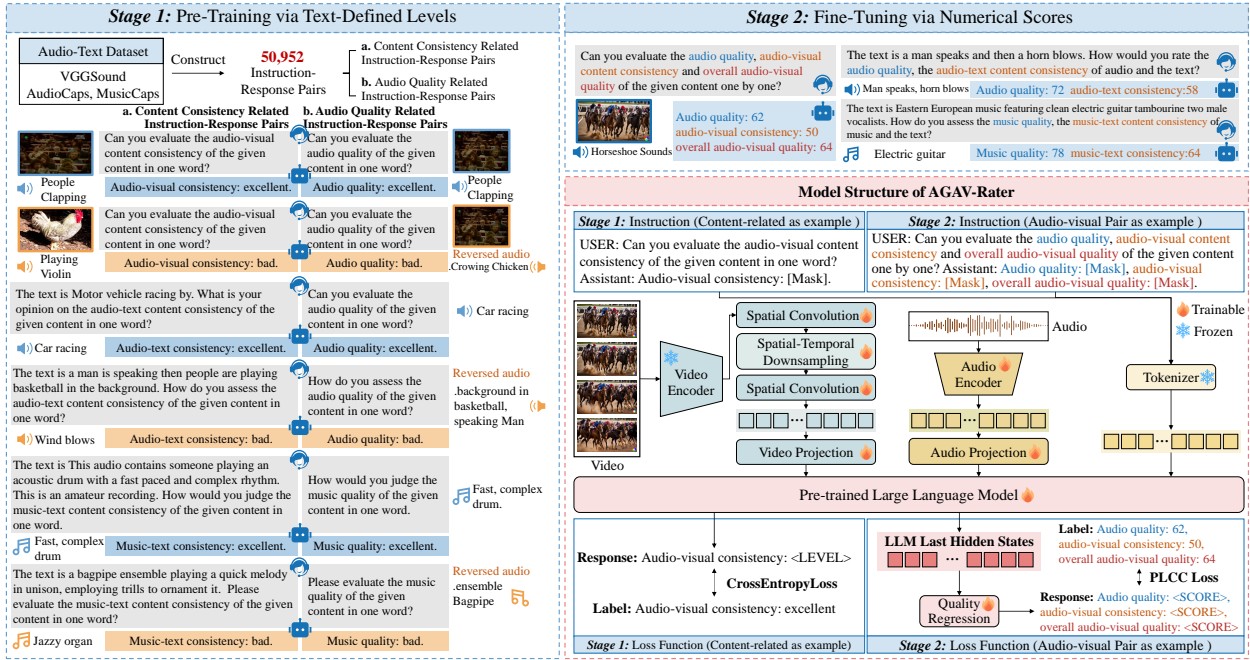

*Figure 3.* Training process and architecture of AGAV-Rater. The training process of AGAV-Rater consists of two steps: first, pre-training AGAV-Rater via text-defined levels, and then fine-tuning it via numerical scores.

## 3.1. AGAVQA-MOS Subset Generation

**AIGC Video Collection.** We first collected high-quality AIGC videos from AIGC video display websites (Sora (sor, 2024), KLing (kli, 2024), and Gen3 (Gen, 2024)) and the video generation benchmark Vbench (Huang et al., 2024). Additionally, we gathered prompts from FETV (Liu et al., 2024b) and generated AIGC videos using closed-source text-to-video platforms (Pika 1.0 (pik, 2023) and Gen3 (Gen, 2024)). From these AIGC videos, we manually selected 386 high-quality videos with clear audio source information, covering 11 types of audio sources. The distribution of main audio sources in AIGC videos is shown in Fig. 2(a).

**Audio Generation from Video.** We utilized 8 latest VTA methods, including Diff-Foley (Luo et al., 2024), Foley-Crafter (Zhang et al., 2024a), VTA-LDM (Hu et al., 2024), Im2wav (Sheffer & Adi, 2023), SpecVQGAN (Iashin & Rahtu, 2021), ModaVerse (Wang et al., 2024f), SVA (Chen et al., 2024), and ElevenLabs (ele, 2023), to generate audio from AIGC videos, thus producing AGAVs. For the Eleven-Labs closed-source platform, we used its API [1], while for the other 7 VTA methods, we used their default weights and code to generate audio. As a result, we obtained a total of 3, 088 AGAVs (8 VTA methods × 386 AIGC videos).

**Human Evaluation.** We asked subjects to rate AGAVs across three dimensions: audio quality, A/V content consistency, and overall A/V quality. Audio quality evaluates the perceived quality of the audio, including clarity, naturalness, and pleasantness. A/V content consistency primarily

assesses whether the audio aligns with the corresponding visual elements in the video. Overall A/V quality evaluates the overall quality of the audio and video, including video quality, audio quality, and the compatibility between audio and video. These three quality dimensions are related yet distinct, offering a comprehensive evaluation of AGAVs from multiple perspectives.

We invited 15 subjects to participate in our subjective experiment. We designed a user interface that allows subjects to watch, listen, and rate the AGAVs across three dimensions. The interface displayed 3 continuous quality rating bars, each labeled with a 1-5 Likert scale for rating. We first explained the experiment requirements and the three rating dimensions to each subject. Then, subjects entered a brief training phase to familiarize themselves with the user interface and scoring rules by watching 24 AGAVs. Afterward, they proceeded to the official testing phase. Finally, we normalized the three-dimensional raw scores to Z-scores ranging from 0 to 100 and calculated the mean Z-scores to obtain the mean opinion scores.

## 3.2. MOSs Analysis

In Fig. 2(c), the maximum, minimum, and average subjective scores for the 8 VTA methods are presented across the three dimensions. We can observe that the AGAVs generated by SVA (Chen et al., 2024) exhibit the best audio quality, which can be attributed to the use of proprietary TTA tools, i.e., AudioGen (Kreuk et al., 2022) and Music-Gen (Copet et al., 2024), to generate high-quality sound

---

[1] https://videotosfx.elevenlabs.io/

effects and background music. ElevenLabs achieves the best A/V content consistency and overall quality by using ChatGPT-4 to extract key sound information from videos and generate high-quality audio with its proprietary TTA tool, ensuring seamless coherence between audio and video. Krippendorff's $\alpha$ (Hayes & Krippendorff, 2007) can be used to measure the quality of the subjects' ratings. We calculate Krippendorff's $\alpha$ for audio quality, A/V content consistency, and A/V overall quality, which are $0.6814$, $0.7343$, and $0.7143$, respectively, indicating appropriate variations among subjects. We also randomly divide subjects into two groups and calculate the SRCC of average scores between the two groups. After ten repetitions, the average SRCC for audio quality, A/V content consistency, and A/V overall quality are $0.8043$, $0.8318$, and $0.8297$, validating rating consistency.

## 3.3. AGAVQA-Pair Subset Collection

Most VTA GitHub pages exhibit superior performance by dubbing the same video using both their approach and other VTA methods. As shown in Fig. 2(b), we collected $294$ publicly available AGAVs from $8$ VTA GitHub pages, including SSV2A (Guo et al., 2024a), ReWaS (Jeong et al., 2025), TIVA (Wang et al., 2024d), V2A-Mapper (Wang et al., 2024a), STAV2A (Ren et al., 2024), V2A-SceneDetector (Yi & Li, 2024), Frieren (Wang et al., 2024g), and SonicVisionLM (Xie et al., 2024). These AGAVs, sourced from third-party platforms, offer a more objective and impartial dataset. These VTA GitHub pages are all released in the past year, representing the latest technology in VTA methods. AGAVs with the same video content are grouped, with the optimal AGAVs already labeled on the GitHub pages. We manually verified the accuracy of the optimal AGAVs, and then formed $75$ instruction-response pairs as follows:

`Instruction`: The video is <video>, Audio 1 is <audio 1>, Audio 2 is <audio 2>, Audio 3 is <audio 3>... Which audio best matches this video in terms of audio content, quality, and rhythm? `Response`: Audio 1.

Due to the inherent subjectivity in human evaluations of AGAVs, different subjects may provide varying scores to the same AGAV. However, when selecting the optimal AGAV, most subjects tend to give the same choice, leading to more reliable quality labels. In some application scenarios, businesses only need to select the highest-quality result from multiple generated options to present to users, without requiring detailed quality scores for each AGAV. Therefore, we construct the AGAVQA-Pair subset. This subset evaluates the performance of AGAV quality assessment methods by measuring their accuracy in selecting the optimal AGAV. More details of the AGAVQA-3k dataset are provided in Appendix A.

## 4. The AGAV-Rater

In this section, we provide a detailed description of the training process and architecture of AGAV-Rater, as illustrated in Fig. 3. We first construct $50,952$ instruction-response pairs for AGAV-Rater pre-training, where no human annotations are required, and two text-defined levels are utilized as labels. Then, we utilize the three-dimensional numerical scores from the AGAVQA-MOS subset as labels to finetune the AGAV-Rater.

## 4.1. Pre-Training via Text-Defined Levels

To alleviate the burden of constructing large-scale quality assessment datasets, we first construct $50,952$ instruction-response pairs from $3$ real-world audio-caption related datasets for AGAV-Rater pre-training, as shown in Fig. 3. The responses are automatically labeled with two text-defined levels (excellent and bad). We select the VGGSound, AudioCaps, and MusicCaps datasets to cover $3$ different scenarios: audio-video, audio-text, and music-text. In the $50,952$ instruction-response pairs, the audio-video, audio-text, and music-text scenarios contain $25,592$, $19,000$, and $6,000$ pairs, respectively. Instruction-response pairs are designed from two perspectives: content consistency and audio quality. Under each scenario, half of the pairs focus on content consistency, and the other half on audio quality. Take the A/V scenario as an example, the instruction for content consistency related pairs is as follows:

*#User:* <audio><video> Can you evaluate the audio-visual content consistency of the given content in one word? *#Assistant:* Audio-visual consistency: `[Mask]`.

In the VGGSound dataset, we consider the A/V content consistency of the original video to be excellent. After replacing the original audio with audio from other categories, the consistency quality becomes bad. In the AudioCaps and MusicCaps dataset, we replace the original caption with another caption and ensure no overlapping nouns between the captions to create audio-text samples with bad audio-text consistency.

Since the unnatural distortions in AIGC audio are difficult to simulate with real-world distortions like white noise or Gaussian noise, we simulate the unnaturalness by reversing the audio. In the VGGSound dataset, the audio quality of the original A/V sample is labeled as excellent. Videos with reversed audio are marked as having bad audio quality. Similarly, for AudioCaps and MusicCaps datasets, if the audio in the audio-caption samples is reversed, the audio quality is labeled as bad. We utilize the content consistency and audio quality related instruction-response pairs together to pre-train the AGAV-Rater, allowing it to develop a basic level of quality perception. The AGAV-Rater utilizes the standard loss function of LMMs, which is the cross-entropy between the labels and output logits.

*Table 1.* Performance comparisons on the AGAVQA-MOS subset from three dimensions. The best performance results are shown in bold, and the second-best performance results are underlined.

| Dimension | | Audio Quality | | | | Content Consistency | | | | Overall Quality | | | |
|---|---|---|---|---|---|---|---|---|---|---|---|---|---|
| Model Type | Model/Metrics | SRCC↑ | KRCC↑ | PLCC↑ | RMSE↓ | SRCC↑ | KRCC↑ | PLCC↑ | RMSE↓ | SRCC↑ | KRCC↑ | PLCC↑ | RMSE↓ |
| Audio-Visual LMMs | PandaGPT (Arxiv 2023) | 0.1326 | 0.0887 | 0.1697 | 10.7643 | 0.2739 | 0.1861 | 0.2943 | 10.8103 | 0.1272 | 0.0844 | 0.1922 | 11.1854 |
| | NextGPT (ICML 2024) | 0.1523 | 0.1029 | 0.0962 | 10.8685 | 0.0076 | 0.0047 | 0.0827 | 11.2758 | 0.0418 | 0.0278 | 0.0576 | 11.3874 |
| | VITA-1.0 (Arxiv 2024) | 0.0717 | 0.0484 | 0.0980 | 10.8600 | 0.0603 | 0.0403 | 0.1015 | 11.2522 | 0.1284 | 0.0859 | 0.1760 | 11.2200 |
| | VideoLLaMA2 (Arxiv 2024) | 0.4079 | 0.2787 | 0.4384 | 9.8146 | 0.2326 | 0.1559 | 0.2706 | 10.8631 | 0.2438 | 0.1652 | 0.2657 | 10.9716 |
| AQA | MOSNet (INTERSPEECH 2019) | 0.4182 | 0.2888 | 0.4926 | 9.5064 | 0.2567 | 0.1713 | 0.2759 | 10.8737 | 0.2723 | 0.1829 | 0.2963 | 10.8916 |
| | STOI-Net (APSIPA ASC 2020) | 0.2071 | 0.1364 | 0.3285 | 10.2281 | 0.0408 | 0.0239 | 0.1280 | 11.2099 | 0.1692 | 0.1179 | 0.2157 | 11.1291 |
| | NISQA (INTERSPEECH 2021) | 0.5258 | 0.3701 | 0.5875 | 8.8416 | 0.3839 | 0.2641 | 0.4064 | 10.3339 | 0.3374 | 0.2286 | 0.3461 | 10.7026 |
| | PAM (INTERSPEECH 2024) | 0.3180 | 0.2149 | 0.3608 | 10.0233 | -0.0183 | -0.0102 | 0.1441 | 11.8682 | 0.1421 | 0.0950 | 0.1876 | 11.3385 |
| Audio-Video Alignment | AVID-CMA (CVPR 2021) | 0.6986 | 0.5101 | 0.7350 | 7.4310 | 0.5486 | 0.3851 | 0.5669 | 9.3384 | 0.6148 | 0.4350 | 0.6246 | 8.8641 |
| | VAST (NIPS 2023) | 0.7640 | 0.5682 | 0.7848 | 6.7285 | 0.6811 | 0.4944 | 0.6958 | 8.1110 | 0.7094 | 0.5166 | 0.7180 | 7.8624 |
| | VALOR (TPAMI 2024) | 0.7474 | 0.5549 | 0.7773 | 6.8629 | 0.6471 | 0.4662 | 0.6635 | 8.4625 | 0.6888 | 0.4975 | 0.7034 | 8.0904 |
| AVQA | DNN-RNT (TIP 2023) | 0.5228 | 0.3656 | 0.5447 | 9.1389 | 0.4348 | 0.2970 | 0.4460 | 10.1231 | 0.4940 | 0.3406 | 0.5031 | 9.8489 |
| | DNN-SND (TIP 2023) | 0.5582 | 0.3932 | 0.5782 | 8.9103 | 0.4686 | 0.3225 | 0.4821 | 9.9072 | 0.5457 | 0.3815 | 0.5548 | 9.4669 |
| | GeneralAVQA (TIP 2023) | 0.6102 | 0.4346 | 0.6458 | 8.3252 | 0.4658 | 0.3219 | 0.4768 | 9.9448 | 0.6007 | 0.4249 | 0.6160 | 8.9777 |
| **AGAV-Rater (Ours)** | | **0.7909** | **0.5980** | **0.8108** | **6.3894** | **0.7553** | **0.5639** | **0.7645** | **7.2956** | **0.7458** | **0.5516** | **0.7552** | **7.4611** |

*Table 2.* Performance comparisons on the TTA and TTM datasets from two dimensions. The best performance results are shown in bold, and the second-best performance results are underlined.

| Dataset | | **Text-to-Audio** | | | | | | **Text-to-Music** | | | | | |
|---|---|---|---|---|---|---|---|---|---|---|---|---|---|
| Dimension | | Audio Quality | | | Content Consistency | | | Audio Quality | | | Content Consistency | | |
| Model Type | Model/Metrics | SRCC↑ | KRCC↑ | PLCC↑ | SRCC↑ | KRCC↑ | PLCC↑ | SRCC↑ | KRCC↑ | PLCC↑ | SRCC↑ | KRCC↑ | PLCC↑ |
| Audio-Visual LMMs | PandaGPT (Arxiv 2023) | 0.0888 | 0.0616 | 0.2049 | 0.1976 | 0.1327 | 0.3436 | 0.1722 | 0.1162 | 0.2560 | 0.0748 | 0.0494 | 0.2240 |
| | NextGPT (ICML 2024) | -0.0160 | -0.0115 | 0.1677 | -0.0097 | -0.0068 | 0.1599 | 0.0498 | 0.0334 | 0.1648 | -0.0586 | -0.0397 | 0.2149 |
| | VITA-1.0 (Arxiv 2024) | 0.1324 | 0.0915 | 0.2559 | -0.0116 | -0.0092 | 0.1521 | -0.0961 | -0.0682 | 0.1526 | 0.1886 | 0.1224 | 0.3325 |
| | VideoLLaMA2 (Arxiv 2024) | 0.4698 | 0.3277 | 0.5061 | 0.5472 | 0.3880 | 0.5514 | 0.5046 | 0.3510 | 0.5258 | 0.1402 | 0.0957 | 0.2754 |
| AQA | MOSNet (INTERSPEECH 2019) | 0.4658 | 0.3328 | 0.4865 | 0.4223 | 0.2987 | 0.4537 | 0.4646 | 0.3110 | 0.4600 | 0.3206 | 0.2213 | 0.3384 |
| | STOI-Net (APSIPA ASC 2020) | 0.4327 | 0.3032 | 0.4603 | 0.4080 | 0.2858 | 0.4425 | 0.2760 | 0.2084 | 0.3346 | 0.2924 | 0.2298 | 0.3734 |
| | NISQA (INTERSPEECH 2021) | 0.4262 | 0.3007 | 0.4603 | 0.4072 | 0.2830 | 0.4353 | 0.6264 | 0.4376 | 0.6394 | 0.5724 | 0.4036 | 0.5859 |
| | PAM (INTERSPEECH 2024) | 0.5165 | 0.3655 | 0.5264 | 0.4100 | 0.2833 | 0.4217 | 0.6435 | 0.4630 | 0.6448 | 0.3465 | 0.2354 | 0.3813 |
| Audio (Music)-Text Alignment | CLAP (ICASSP 2023) | 0.7040 | 0.5231 | 0.7149 | 0.6700 | 0.4876 | 0.6782 | 0.8103 | 0.6265 | 0.8096 | 0.7474 | 0.5553 | 0.7445 |
| | TTM-Retrieval (ICASSP 2023) | 0.6121 | 0.4465 | 0.6455 | 0.5818 | 0.4173 | 0.5995 | 0.7586 | 0.5649 | 0.7649 | 0.6974 | 0.5102 | 0.7063 |
| | VAST (NIPS 2023) | 0.7255 | 0.5421 | 0.7312 | 0.6879 | 0.5021 | 0.6956 | 0.8289 | 0.6412 | 0.8175 | 0.7441 | 0.5557 | 0.7494 |
| | VALOR (TPAMI 2024) | 0.6711 | 0.4972 | 0.6879 | 0.5948 | 0.4250 | 0.5963 | 0.2619 | 0.1838 | 0.3146 | 0.2350 | 0.1653 | 0.3013 |
| **AGAV-Rater (Ours)** | | **0.7390** | **0.5566** | **0.7495** | **0.7330** | **0.5427** | **0.7367** | **0.8322** | **0.6500** | **0.8277** | **0.7719** | **0.5819** | **0.7811** |

## 4.2. Fine-Tuning via Numerical Scores

**Multi-Dimensional Evaluation Instruction.** During the subjective experiment, we observe that subjects typically first evaluate audio quality, followed by A/V content consistency, and finally the overall A/V quality. To mimic the human thought process, we design a multi-dimensional instruction to fine-tune the LMM to evaluate the three dimensions sequentially, e.g.,

*#User:* <audio><video>Can you evaluate the audio quality, audio-visual content consistency, and overall audio-visual quality of the given content one by one? *#Assistant:* Audio quality: [Mask], audio-visual consistency: [Mask], overall audio-visual quality: [Mask].

**Masking Quality Levels.** To further improve the model's evaluation capability, we mask the quality levels in instructions, which can prevent the model from overly relying on the ground truth of other scores when predicting quality. It can encourage the model to utilize the inferred quality levels to guide subsequent score predictions, thereby deepening the understanding of the relationships between 3 quality dimensions.

**Numerical Score Prediction.** Most researchers (Wu et al.; Kou et al., 2024; Zhang et al., 2024b) consider it suboptimal to directly tune LMMs to output numerical scores. They con-

vert the MOSs into five text-defined rating levels (excellent, good, fair, poor, and bad) and subsequently format them into instruction-response pairs for visual instruction tuning. However, converting MOSs to five rating levels discards a significant amount of information. We directly regress the LMM's last hidden states to output three-dimensional numerical scores. The loss between the predicted scores and the ground truth is calculated using PLCC Loss. By predicting numerical scores instead of text-defined levels, AGAV-Rater can more precisely understand human subjective perception.

## 4.3. Model Structure

As shown in Fig. 3, the structure of the AGAV-Rater is based on the recently released open-source LMM, VideoLLaMA2 (Cheng et al., 2024), which exhibits excellent A/V perception abilities and strong language comprehension. The video is first converted into a 1fps image sequence. Then, the image sequence and audio signal are separately encoded by the video encoder and audio encoder. After encoding, they are projected into the same vector space through the video projection and audio projection, and input into the large language model together with text embedding. It enables us to process AGAVs, audio-text, and music-text quality assessment tasks within a unified model framework.

*Table 3.* Answer accuracy on AGAVQA-Pair subset. Based on the VTA method source, we divide the AGAVQA-Pair subset into 8 categories, with **All** representing the entire dataset. **AGAV-Rater** is trained on the AGAVQA-MOS subset, demonstrating cross-dataset performance on the AGAVQA-Pair subset. The best result is shown in bold, and the second-best is underlined.

| Category | SonicVisionLM | Frieren | V2AMapper | TIVA | V2A-SceneDetector | STAV2A | SSV2A | ReWaS | All |
|---|---|---|---|---|---|---|---|---|---|
| Random | 0.33 | 0.20 | 0.41 | 0.20 | 0.50 | 0.25 | 0.20 | 0.25 | 0.32 |
| *Question Type*: Multi-Input Comparison, *w/o* fine-tuning on the AGAVQA-MOS subset. | | | | | | | | | |
| PandaGPT (Su et al., 2023) | 0.29 | 0.20 | 0.43 | 0.22 | 0.43 | 0.23 | 0.20 | 0.22 | 0.26 |
| NextGPT (Wu et al., 2024) | 0.33 | 0.20 | 0.28 | 0.04 | 0.36 | 0.30 | 0.16 | 0.22 | 0.22 |
| VITA-1.0 (Fu et al., 2024) | 0.33 | 0.12 | 0.26 | 0.08 | 0.43 | 0.18 | 0.04 | 0.19 | 0.17 |
| VideoLLaMA2 (Cheng et al., 2024) | 0.38 | 0.17 | 0.26 | 0.14 | 0.21 | 0.18 | 0.12 | 0.28 | 0.21 |
| Gemini-1.5 Flash-8b (Team et al., 2024a) | 0.10 | 0.05 | 0.20 | 0.06 | 0.29 | 0.10 | 0.04 | 0.17 | 0.11 |
| Gemini-1.5 Pro (Team et al., 2024a) | 0.29 | 0.23 | 0.35 | 0.22 | 0.07 | 0.13 | 0.28 | 0.17 | 0.23 |
| Gemini-2.0 Flash (Team et al., 2024a) | 0.10 | 0.18 | 0.43 | 0.26 | 0.64 | 0.25 | 0.24 | 0.19 | 0.27 |
| Reka Core (Team et al., 2024b) | 0.19 | 0.18 | 0.39 | 0.22 | 0.43 | 0.18 | 0.08 | 0.22 | 0.23 |
| Reka Flash (Team et al., 2024b) | 0.24 | 0.20 | 0.30 | 0.30 | 0.50 | 0.20 | 0.28 | 0.22 | 0.26 |
| GPT-4o+Audio Caption (Hurst et al., 2024) | 0.29 | 0.20 | 0.46 | 0.18 | 0.57 | 0.30 | 0.20 | 0.36 | 0.29 |
| GPT-4o+Video Caption (Hurst et al., 2024) | 0.29 | 0.18 | 0.48 | 0.16 | 0.64 | 0.40 | 0.12 | 0.36 | 0.30 |
| *Question Type*: Single-input Scoring, *w/o* fine-tuning on the AGAVQA-MOS subset. | | | | | | | | | |
| PandaGPT (Su et al., 2023) | 0.29 | 0.17 | 0.67 | 0.10 | 0.57 | 0.20 | 0.20 | 0.33 | 0.35 |
| NextGPT (Wu et al., 2024) | 0.43 | 0.08 | 0.28 | 0.30 | 0.57 | 0.30 | 0.40 | 0.33 | 0.31 |
| VITA (Fu et al., 2024) | 0.50 | 0.17 | 0.39 | 0.10 | 0.29 | 0.30 | 0.20 | 0.22 | 0.27 |
| VideoLLaMA2 (Cheng et al., 2024) | 0.71 | 0.25 | 0.67 | 0.30 | 0.29 | 0.10 | 0.20 | 0.44 | 0.40 |
| *Question Type*: Single-input Scoring, *with* fine-tuning on the AGAVQA-MOS subset. | | | | | | | | | |
| AVID-CMA (Morgado et al., 2021) | 0.29 | 0.58 | 0.61 | 0.50 | **0.71** | 0.50 | 0.40 | 0.44 | 0.52 |
| VALOR (Liu et al., 2024a) | **1.00** | 0.75 | 0.72 | 0.70 | **0.71** | **0.70** | 0.40 | 0.44 | 0.55 |
| VAST (Chen et al., 2023) | 0.86 | 0.83 | 0.78 | **0.80** | 0.43 | 0.40 | 0.40 | **0.56** | 0.64 |
| **AGAV-Rater** (Ours) | **1.00** | **0.92** | **0.83** | **0.80** | **0.71** | **0.70** | **0.60** | **0.56** | **0.78** |

## 5. Experiments

### 5.1. Experimental Settings

In this paper, we fine-tune the AGAV-Rater from the pre-trained weights of VideoLLaMA2 (Cheng et al., 2024). The AGAV-Rater model is implemented with PyTorch and trained on two 96GB H20 GPUs. The learning rate is set to $1e - 5$, and the batch size is set to 9. During pre-training, the number of training epochs is set to 1, and optimization is performed. For fine-tuning, the number of training epochs is set to 5 on the AGAVQA-MOS subset and 10 on the TTA and TTM datasets (Deshmukh et al., 2024). Fine-tuning the AGAV-Rater model on the AGAVQA-MOS subset for 5 epochs using two 96GB H20 GPUs takes approximately 5 hours. All experiments for each method are retrained on the AGAVQA-MOS subset using 5-fold cross-validation. The reported performance of the AGAV-Rater is evaluated on the final weights after training.

### 5.2. Compared Methods

Since no specific method has been proposed for evaluating AGAVs, we select state-of-the-art methods from four areas for comparison: audio-visual LMMs, AQA, AVQA, and multimodal alignment, including:

- Audio-visual LMMs: PandaGPT (Su et al., 2023), NextGPT (Wu et al., 2024), VITA-1.0 (Fu et al., 2024), and VideoLLaMA2 (Cheng et al., 2024).
- AQA: MOSNet (Lo et al., 2019), STOI-Net (Zezario et al., 2020), NISQA (Mittag et al., 2021), and PAM

(Deshmukh et al., 2024).

- AVQA: DNN-RNT (Cao et al., 2023a), DNN-SND (Cao et al., 2023a), and GeneralAVQA (Cao et al., 2023b).
- Multimodal alignment: AVID-CMA (Morgado et al., 2021) aligns video features with audio features. VAST (Chen et al., 2023) and VALOR (Liu et al., 2024a) map video, audio, and text into the same semantic space. CLAP (Elizalde et al., 2023) and TTM-Retrieval (Doh et al., 2023) align audio and music with text, respectively.

Except for audio-visual LMMs, all methods are retrained on the AGAVQA-MOS, TTA, and TTM datasets after loading the default weights. Original multimodal alignment methods extract audio and video features using their encoders, then align them into a common vector space. We load the default parameters and use these encoders to extract audio and video features, which are then concatenated. The concatenated features are subsequently fed into a fully connected layer with an output dimension of 3 to predict the three-dimensional scores.

Audio-visual LMMs are directly tested on the dataset using their default weights. For quality-related questions, most audio-visual LMMs respond with text-defined quality levels and are unable to stably output numerical scores. Therefore, following the testing scheme designed in (Wu et al.), we prompt LMMs to receive quality level tokens. Then, extract the probability distribution $\mathcal{X}$ of the predicted level token and convert it into the final predicted scores $S_{LMM}$ as

follows:

$$S_{\text{LMM}} = \sum_{i=1}^{5} i \times \frac{e^{\mathcal{X}_{l_i}}}{\sum_{j=1}^{5} e^{\mathcal{X}_{l_j}}} \qquad (1)$$

where $\{l_i|_{i=1}^{5}\} = \{excellent, good, fair, poor, bad\}$ are the standard text quality levels. A detailed introduction to compared methods can be found in Appendix C.3.

## 5.3. Evaluation on Multi-dimensional Scoring Tasks

We test the multi-dimensional scoring ability of AGAV-Rater on the AGAVQA-MOS, TTA, and TTM datasets, using Spearman Rank-order Correlation (SRCC), Kendall Rank-order Correlation (KRCC), Pearson Linear Correlation (PLCC), and Root Mean Squared Error (RMSE). The experimental results are presented in Tab. 1 and Tab. 2. AGAV-Rater demonstrates the best performance across all three datasets. Especially in the content consistency dimension, the SRCC metric shows 11%, 7%, and 3% improvements on the AGAVQA-MOS, TTA, and TTM datasets, respectively. This highlights the powerful content understanding capability of LMMs, which can significantly aid in evaluating the consistency of AIGC audio and video (text). Furthermore, the poor performance of audio-visual LMMs shows that they can not adapt well to quality assessment tasks, proving that our training process effectively enhances the LMM's understanding of human perception. Traditional AQA and AVQA methods perform weaker on the A/V content consistency compared to the audio quality, as these methods have difficulty adapting to the unique distortions in AIGC media. Audio-video alignment methods focus on the semantic alignment between audio and video, being capable of recognizing semantic-level distortions in AGAVs and demonstrating suboptimal performance.

## 5.4. Evaluation on Optimal AGAV Selection Tasks

The AGAVQA-Pair subset was collected from 8 VTA webpages, dividing it into 8 corresponding categories. For each category, the optimal AGAV is generated by the corresponding VTA method. Since there is no overlap between the video content in the AGAVQA-MOS and AGAVQA-Pair subsets, we perform cross-dataset validation experiments by training the AGAV-Rater on the AGAVQA-MOS subset and testing on the unseen AGAVQA-Pair subset. We collect 11 audio-visual LMMs as comparison methods, including 4 open-source LMMs and 7 closed-source LMMs. For GPT-4o+audio caption, we utilize GPT-4o-audio to generate captions for audio, then feed the text, video, and audio captions into GPT-4o. Similarly, for GPT-4o+video caption, we utilize GPT-4o to generate captions for video, then feed the text, audio, and video captions into GPT-4o-audio. As shown in Tab. 3, we test their accuracy in answering optimal AGAV questions on the AGAVQA-Pair subset. We designed two types of instructions to prompt LMMs for

the optimal AGAV. For multi-input comparisons instruction type, we utilize instruction-response pairs in the AGAVQA-Pair subset to let LMMs answer the optimal audio number. To prevent the model from guessing, we shuffle the audio order and iterate through each audio number as the correct answer, inputting them into LMMs to calculate the response accuracy. This instruction type applies to both open-source and closed-source LMMs, with optimal AGAV judgment based on their textual responses. For the single-input scoring instruction type, we sequentially ask the overall quality of each AGAV and compute the final predicted scores by Eq. 1. The AGAV with the highest score is selected as the optimal AGAV. Since closed-source LMMs cannot access the token probability distribution, this method is only applied to open-source LMMs. The above 11 audio-visual LMMs use their original model parameters without fine-tuning on the AGAVQA-MOS subset. We also test the accuracy of the audio-video alignment methods which have been fine-tuned on the AGAVQA-MOS subset. As seen in Tab. 3, our method achieves the highest accuracy across all 8 VTA methods, demonstrating that our model has the best ability to judge the optimal AGAV and exhibits strong transferability.

## 5.5. Ablation Study

**Effects of Pretraining Procedure.** The "*w/o* Pretrain" column in Tab. 4 shows AGAV-Rater's performance without the pretraining step. Compared to the AGAVQA-MOS subset, the pretraining step significantly enhances AGAV-Rater's performance on the smaller TTA and TTM datasets. This is because pretraining provides rich prior knowledge, which effectively compensates for the limited scale and diversity of these smaller datasets. As shown in Tab. 1, VideoLLaMA2 exhibits relatively weak performance on the content consistency dimension for the TTM dataset. The pre-training procedure, through the music-text instruction-response pairs, enhances the music perception ability of AGAV-Rater, thereby improving its performance on the consistency dimension of the TTM dataset. On the AGAVQA-MOS subset, the pretraining step shows a more significant improvement in the overall A/V quality, compared to audio quality and A/V content consistency. We believe this is because AGAV-Rater relies on the scores of audio quality and A/V content consistency when predicting the overall A/V quality, and pretraining helps AGAV-Rater learn audio quality and A/V content consistency faster, reducing disturbances in learning overall A/V quality.

**Effects of Scoring Method.** We convert MOSs into text-defined levels and use text-defined levels as labels in the fine-tuning step to replace MOSs. As shown in the "Finetuning with Levels" column of Tab. 4, replacing numerical scores with text-defined levels for fine-tuning results in a slight performance decrease. This demonstrates that the AGAV-

Table 4. Ablation study of the proposed AGAV-Rater.

| Dataset | AGAVQA-MOS | | | | | | TTA | | | | | | TTM | | | | | |
|---|---|---|---|---|---|---|---|---|---|---|---|---|---|---|---|---|---|---|
| Dimension | Audio Quality | | Consistency | | Overall Quality | | Audio Quality | | Consistency | | | | Audio Quality | | Consistency | | | |
| Strategy | SRCC ↑ | PLCC ↑ | SRCC ↑ | PLCC ↑ | SRCC ↑ | PLCC ↑ | SRCC ↑ | PLCC ↑ | SRCC ↑ | PLCC ↑ | | | SRCC ↑ | PLCC ↑ | SRCC ↑ | PLCC ↑ | | |
| *w/o* Pretrain | 0.7856 | 0.8030 | 0.7503 | 0.7599 | 0.7141 | 0.7202 | 0.7040 | 0.7117 | 0.7104 | 0.7215 | | | 0.8218 | 0.8233 | 0.7042 | 0.7188 | | |
| Finetuning with Levels | 0.7832 | 0.8043 | 0.7499 | 0.7451 | 0.7078 | 0.7167 | 0.7005 | 0.7051 | 0.7249 | 0.7324 | | | 0.8102 | 0.8121 | 0.7359 | 0.7532 | | |
| Single-Dimension Instruction | 0.7845 | 0.8035 | 0.7511 | 0.7599 | 0.6865 | 0.7007 | 0.7143 | 0.7152 | 0.7286 | 0.7277 | | | 0.8297 | 0.8253 | 0.7565 | 0.7759 | | |
| **AGAV-Rater** | **0.7909** | **0.8108** | **0.7553** | **0.7645** | **0.7458** | **0.7552** | **0.7390** | **0.7495** | **0.7330** | **0.7367** | | | **0.8322** | **0.8277** | **0.7719** | **0.7811** | | |

Table 5. Ablation study of the base models.

| Dimension | Audio Quality | | Consistency | | Overall Quality | |
|---|---|---|---|---|---|---|
| Metric | SRCC ↑ | PLCC ↑ | SRCC ↑ | PLCC ↑ | SRCC ↑ | PLCC ↑ |
| GroundingGPT (ACL 2024) | 0.4387 | 0.4494 | 0.5067 | 0.4764 | 0.4975 | 0.5297 |
| OneLLM (CVPR 2024) | 0.6578 | 0.6879 | 0.6184 | 0.6297 | 0.6327 | 0.6388 |
| **AGAV-Rater** | **0.7909** | **0.8108** | **0.7553** | **0.7645** | **0.7458** | **0.7552** |

Rater, by using quality regression to map hidden states to numerical scores, allows the AGAV-Rater to more finely learn human subjective perception.

**Effects of Multi-Dimension Instruction.** To guide LMMs to score according to the human thought process, we design a multi-dimensional instruction that enables AGAV-Rater to predict scores for all three dimensions simultaneously. For comparison experiments, we break this down into three single-dimensional instructions, where each instruction focuses on one dimension's quality. As shown in the "Single-Dimension Instruction" column of Tab 4, we can see that the multi-dimensional instruction improves AGAV-Rater's performance on the overall A/V quality. This is because guiding AGAV-Rater to first consider audio quality and A/V content consistency helps better predict overall A/V quality. Additionally, the multi-dimensional instruction enables mutual enhancement between the two dimensions on the TTA and TTM datasets, further boosting performance.

**Effects of the Base Models.** We conduct ablation studies using GroundingGPT (Li et al., 2024) and OneLLM (Han et al., 2024) as base models on the AGAVQA-MOS subset. For OneLLM and GroundingGPT, we first load the default weights, and then fine-tune them using the official training code on the AGAVQA-MOS subset. To ensure fairness, we also add the quality regression module, directly regressing the LLM's last hidden states to output three-dimensional numerical scores. As shown in Tab. 5, AGAV-Rater achieves the best performance. The main reason for this is that VideoLLaMA2 is designed for audio-video content understanding and pre-trained on more diverse audio-video datasets, making it more suitable for our quality assessment task. GroundingGPT focuses more on localization and visual understanding and is not designed or trained to understand continuous audio-video content. Its ability to comprehend video quality may be weaker. OneLLM is a general multimodal model that, while supporting audio and video processing, is not specifically optimized or enhanced for video and audio alignment. Its audio-related dataset only includes audio-text data, and OneLLM is more suited to text-vision or text-audio matching and understanding, rather than specific audio-video content.

### 5.6. Enhancing ElevenLabs Results via AGAV-Rater

We finally conduct a subjective experiment to demonstrate that AGAV-Rater can help ElevenLabs select the high-quality audio to present to users. We collect 230 silent AIGC videos from T2V-CompBench (Ji et al., 2024) and Sora (sor, 2024), generating 2 AGAVs for each video using ElevenLabs. We then utilize AGAV-Rater to select the higher-quality AGAVs. A total of 10 subjects are invited to watch and listen to AGAVs. 80% of them prefer the AGAVs selected by AGAV-Rater for its better quality. This demonstrates that AGAV-Rater can be applied in real-world scenarios to help improve the quality of outputs from VTA methods. On the project page[2], we display the AGAVs that AGAV-Rater considers high quality and low quality.

## 6. Conclusion

In this paper, we construct AGAVQA-3k, the first AGAV quality assessment dataset, which labels AGAV quality in two ways: multi-dimensional score prediction and optimal AGAV selection. We propose a novel LMM-based AGAV quality assessment method, AGAV-Rater. AGAV-Rater demonstrates superior score prediction capabilities on the AGAVQA-MOS, TTA, and TTM datasets, and achieves the highest accuracy in identifying the optimal AGAV on the AGAVQA-Pair subset. AGAV-Rater enhances users a better audio-visual experience and enhance the quality of VTA method outputs.

## Impact Statement

The quality of AI-generated audio and video must align with human preferences. Among existing models, AGAV-Rater achieves the highest consistency with human perceptual evaluations of AGAVs, indicating its potential for supervising and controlling the quality of AGAVs. We will continue to focus on and advance this research in the future.

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

*Table 6.* Categorization of prompts for generating AIGC videos.

| Category | Description | Key Words |
|---|---|---|
| Animals | Describe animal behaviors and movements | birds fly, dogs sleep, dogs sneeze, penguins walk, fishes float, bees fly, ducks swim, turtles swim, dragons fly, camels walk, cats meow. |
| Artificial | Include environmental sounds generated by human-made objects | Flag waves, country road, restaurant, rocket goes off, fireworks explode, burn a candle, city road, fire. |
| Food | Describe food preparation and eating | pour whiskey, pack box, bread, coffee machine, make a salad, preparation of vegetable soup, sparkling champagne, cook pasta coffee beans fall, pour in a cocktail glass. |
| People | Describe human action | cook, water ski, go boating, wash gravel, swim, walk, get out a lake, box, pour liquid, fight with swords. |
| Nature | Include natural environmental sounds | flood water, ice melt, fallen leaves, water plants, mountain river, forest, wind, beach, sea, snow all, waterfall, rain. |
| Vehicles | Describe vehicles | car, war vehicle, boat, aircraft, bike. |

# A. More Details of AGAVQA-3k Construction.

## A.1. Detailed Information of AIGC Video Collection

We begin by collecting AIGC videos from public display websites, including Sora[3], KLing[4], and Gen3[5]. Additionally, we gather AIGC videos generated by AnimateDiff (Guo et al., 2024b), CogVideo (Hong et al., 2022), Gen3 (Gen, 2024), KLing (kli, 2024), LaVie (Wang et al., 2023), Pika (pik, 2023), and TF-T2V (Wang et al., 2024e) from the video generation benchmark Vbench (Huang et al., 2024). To diversify the types of audio sources in the videos, we also collect prompts containing audio information from FETV (Liu et al., 2024b). As shown in Tab. 6, these prompts are categorized into 6 types: animals, artificial, food, people, nature, and vehicles. We also list the key terms associated with each category. Finally, we use these prompts to generate AIGC videos via closed-source text-to-video platforms (Pika 1.0 (pik, 2023) and Gen3 (Gen, 2024)).

## A.2. Detailed Information of VTA Methods

We utilize 16 state-of-the-art VTA methods to construct the AGAVQA-3k dataset, encompassing both open-source and closed-source methods. For open-source models, we rely on official repositories and utilize default weights to generate audio. For closed-source models, we utilize publicly accessible APIs provided by open platforms. For models without publicly available code, we collect public AGAVs showcased on their GitHub pages.

### A.2.1. DIFFUSION BASED VTA METHODS

**Diff-Foley.**: Diff-Foley (Luo et al., 2024) is a synchronized video-to-audio synthesis method that utilizes a latent diffusion model (LDM) to generate high-quality audio with improved temporal synchronization and audio-visual relevance.

**FoleyCrafter.** FoleyCrafter (Zhang et al., 2024a) is a pluggable module integrated into a text-to-audio generator, enabling it to generate high-quality audio synchronized with video content. It primarily utilizes two key components: a semantic adapter for semantic alignment and a temporal controller for temporal synchronization.

**VTA-LDM.** VTA-LDM (Hu et al., 2024) leverages the recently popular grounding segment anything model (Grounding SAM) to extract fine-grained semantic features from video frames and then uses a LDM to generate high-quality audio.

**SSV2A.** SSV2A[6] (Guo et al., 2024a) is a Sound Source-Aware Video-to-Audio generator, which can locally perceive multimodal sound sources from a scene through visual detection and cross-modality translation.

**ReWaS.** ReWaS (Jeong et al., 2025) is a video-and-text-to-sound generation method, where video conditions control the text-to-audio generation model to create audio that matches the video.

**TIVA.** TIVA[7] (Wang et al., 2024d) is a novel time-aligned video-to-audio generator that jointly achieves semantic matching and temporal synchronization when generating audio. TIVA encodes the semantic information of the video and predicts its

---

[3]https://openai.com/sora/

[4]https://klingai.com/

[5]https://runwayml.com/research/introducing-gen-3-alpha

[6]https://ssv2a.github.io/SSV2A-demo/

[7]https://tiva2024.github.io/TiVA.github.io/home/

rhythmic layout, then utilizes this information as conditioning for a latent diffusion-based audio generator to produce the audio.

**V2A-Mapper.** V2A-Mapper[8] (Wang et al., 2024a) employs CLIP, CLAP, and AudioLDM to design a lightweight VTA method. It maps the latent space from the visual CLIP model to the auditory CLAP model and then uses the pre-trained AudioLDM to generate high-fidelity, visually-aligned sound.

**STAV2A.** STAV2A[9] (Ren et al., 2024) is a semantic and temporal aligned video-to-audio method that generates audio by conditioning on both text and video features. STAV2A utilizes an LDM initialized with text-to-audio prior knowledge and guided by cross-modal features from both text and video.

**V2A-SceneDetector.** V2A-SceneDetector[10] (Yi & Li, 2024) combines LDM with a scene detector to address the challenge of multiple visual scene transitions in videos. It can identify and handle multiple scenes in a video, generating corresponding audio for each.

### A.2.2. TRANSFORMER BASED VTA METHODS

**Im2wav.** Im2wav (Sheffer & Adi, 2023) is based on two transformer language models. First, a language model generates a low-level audio representation. Then, an additional language model upsamples the audio tokens to generate high-fidelity audio samples.

**SpecVQGAN.** SpecVQGAN (Iashin & Rahtu, 2021) is a visually-induced audio generation method. It utilizes a transformer to sample a new spectrogram from a pre-trained spectrogram codebook, given the set of video features, thereby generating the corresponding audio.

### A.2.3. LLM BASED VTA METHODS

**ModaVerse.** ModaVerse (Wang et al., 2024f) is a multi-modal large language model capable of understanding and converting content across various modalities, including images, videos, and audio. We leverage its video-to-audio capability to add sound to silent videos.

**SVA.** SVA (Chen et al., 2024) is a semantically consistent video-to-audio generation framework. SVA leverages the capabilities of multi-modal large language models (MLLMs) to understand the video semantics from key frames and generate creative audio plans. These plans are then used as prompts for text-to-audio models, simultaneously generating sound effects and background music.

**SonicVisionLM.** SonicVisionLM[11] (Xie et al., 2024) is a new framework designed to generate a wide range of sound effects by leveraging vision-language models (VLMs). It uses VLMs to recognize events in the video and generate sounds that match the video content.

### A.2.4. FLOW MATHCING BASED VTA METHOD

**Frieren.** Frieren[12] (Wang et al., 2024g) is a VTA model based on rectified flow matching. Frieren generates high-quality audio in just a few, or even a single, sampling step through backflow and a guided vector field distillation process.

### A.2.5. PROPRIETARY VTA METHOD

**ElevenLabs.** ElevenLabs[13] (ele, 2023) utilizes the ElevenLabs texts to sound effects API. It extracts sound source information from key frames using ChatGPT-4o and then inputs this data into ElevenLabs to generate the corresponding audio.

---

[8]https://v2a-mapper.github.io/
[9]https://y-ren16.github.io/STAV2A/
[10]https://1mageyi.github.io/V2A-SceneDetector.demo/
[11]https://yusiissy.github.io/SonicVisionLM.github.io/
[12]https://frieren-v2a.github.io/
[13]https://videotosfx.elevenlabs.io/

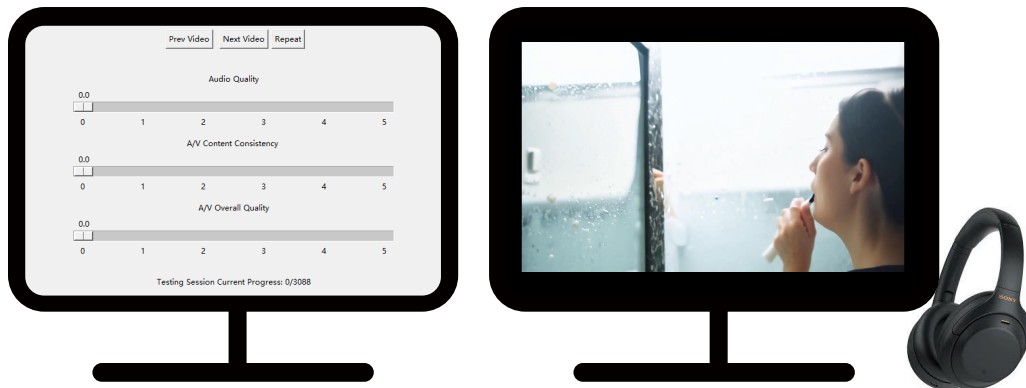

*Figure 4.* An example of the scoring interface. The audio-visual setup consists of two 1080p monitors and a headphone. One monitor displays the scoring interface, while the other is used for subjects to watch the AGAV videos and listen to the audio through the headphones.

## B. More Details of Human Evaluation

### B.1. Scoring Dimensions

During the subjective scoring process, participants are asked to evaluate the AGAVs from three dimensions: audio quality, A/V content consistency, and overall A/V quality. These three dimensions provide a comprehensive and detailed assessment of the AGAVs. Audio quality mainly focuses on the naturalness and clarity of the audio, minimizing the influence of the video content. Higher scores indicate that the audio is more natural, clear, and free of distortion, while lower scores reflect that distortions and noise in the sound degrade the listener's auditory experience. The scoring range is from 0 to 5, with scores accurate to one decimal place. The scoring criteria are as follows:

- 4–5 (Excellent): The audio is natural, clear, and free of distortion.

- 3–4 (Good): The audio is generally clear but contains slight distortion or noise.

- 2–3 (Fair): The audio is somewhat muffled, with noticeable distortion or noise.

- 1–2 (Poor): The audio is quite muffled, with severe distortion and noise that significantly affect the listening experience.

- 0–1 (Bad): The audio is completely distorted.

A/V content consistency is independent of audio quality and primarily focuses on whether the sounds or music in the audio align with the video content. Higher scores indicate a strong correlation between the audio and the video content, while lower scores indicate a low correlation, with the audio lacking the sound elements intended to be conveyed by the video. The scoring criteria are as follows:

- 4–5 (Excellent): The audio content is highly consistent with the video content.

- 3–4 (Good): The audio content is generally consistent with the video content.

- 2–3 (Fair): The audio content has limited correlation with the video content.

- 1–2 (Poor): The audio content is mostly inconsistent with the video content.

- 0–1 (Bad): The audio is largely distorted, and the audio content is completely inconsistent with the video content.

Overall A/V quality mainly focuses on the overall perceptual experience of the audio and video, including audio quality, A/V content consistency, and A/V temporal synchronization. Higher scores indicate clear and natural audio, with both content and timing highly consistent with the video. Lower scores indicate obvious audio distortion and low correlation with the video content. The scoring criteria are as follows:

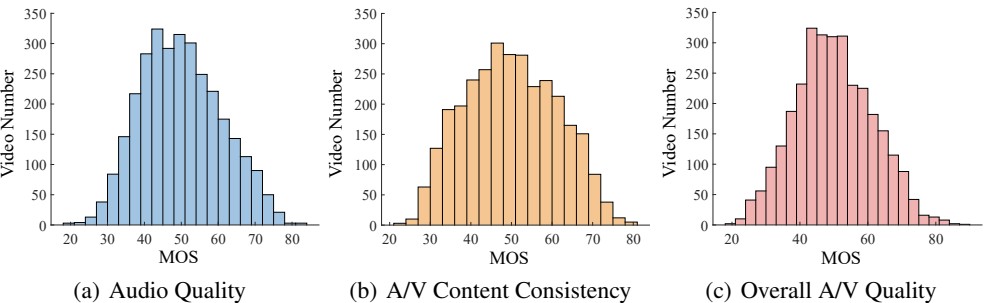

(a) Audio Quality  (b) A/V Content Consistency  (c) Overall A/V Quality

*Figure 5.* Distribution of MOSs across three dimensions in the AGAVQA-MOS subset.

- 4–5 (Excellent): The overall quality of the audio and video is excellent.

- 3–4 (Good): The overall quality of the audio and video is good, with slight distortion.

- 2–3 (Fair): The overall quality of the audio and video is average, with noticeable distortion.

- 1–2 (Poor): The overall audio and video quality is poor, with severe distortion or mostly inconsistent content.

- 0–1 (Bad): The overall audio and video quality is very poor, with completely inconsistent content and severely distorted audio or video.

We developed a subjective evaluation guideline that includes the scoring criteria for the three dimensions mentioned above. Before each participant begins the subjective experiment, we guide them through reading this document to familiarize themselves with the scoring criteria for each dimension, thereby ensuring accuracy and consistency of the ratings across different participants.

### B.2. Scoring Interface

The example of the scoring interface is shown in Fig. 4. We designed the scoring interface using the Python Tkinter package. Before scoring, each subject is prompted to enter their username, which allows the retrieval of their scoring progress and subsequent presentation of the scoring interface. The interface includes three continuous quality rating bars, three navigation buttons, and a display of the number of AGAVs that have been scored. Each rating bar is labeled with a 1–5 Likert scale, and participants can drag the slider to assign scores accurate to one decimal place. The navigation buttons, including "Prev", "Repeat", and "Next", enable participants to modify the previous AGAV's score, replay the current AGAV, and submit their score to proceed to the next AGAV, respectively, facilitating efficient scoring. All AGAVs are viewed at their original resolution without any scaling or cropping.

### B.3. Evaluation Environment

The official testing phase was conducted in a controlled laboratory setting with normal indoor lighting and a quiet environment. Subjects were seated at a comfortable distance of approximately 60 cm from the screen. We used a Redmi 23.8-inch monitor with a resolution of $1920 \times 1080$ and Sony WH-1000XM4 headphones to minimize distortions introduced by the viewing and listening devices during human evaluation.

### B.4. Subject Selection

We invited subjects familiar with AVQA and AGAV to participate in on-site training sessions. Detailed explanations of the scoring criteria for each dimension were provided, along with additional AGAV samples for practice. Expert reviewers then evaluated the annotations and selected 15 qualified subjects. Each subject rated all $3,088$ samples in the AGAVQA-MOS subset in a randomized order. To prevent fatigue, each subject was limited to rating a maximum of 60 samples per day, completing the entire task in approximately two months. This protocol ensured the validity and robustness of each subject's ratings.

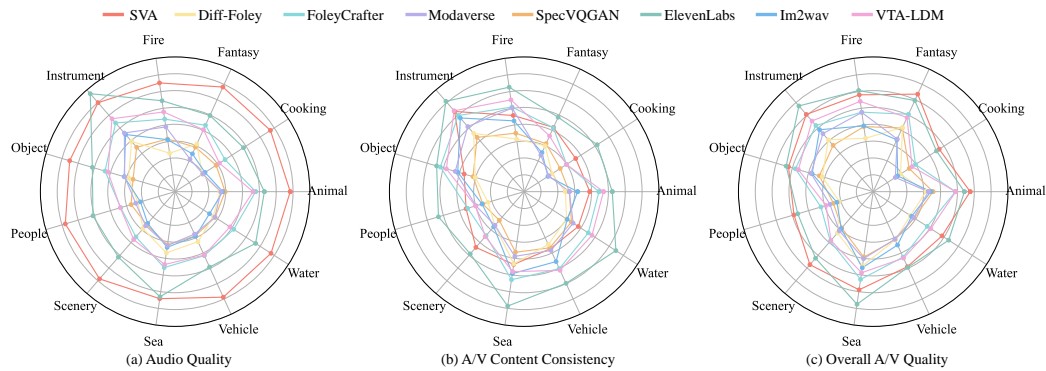

*Figure 6.* Comparison of mean MOSs of different VTA methods across various audio sources in videos. (a) Results on audio quality, (b) Results on A/V content consistency, (c) Results on overall A/V quality.

*Table 7.* Standard deviation of subjective scores for each category in the AGAVQA-MOS subset.

| Animal | Water | People | Vehicle | Object | Scenery | Sea | Fantasy | Fire | Instrument | Cooking |
|--------|-------|--------|---------|--------|---------|------|---------|------|-----------|---------|
| 12.56 | 12.16 | 12.20 | 11.79 | 12.57 | 12.74 | 12.37 | **13.11** | 12.35 | 11.93 | 12.23 |

### B.5. More Subjective Scores Analysis

In the AGAVQA-MOS subset, we collected MOSs across three dimensions. SRCC between audio quality and A/V content consistency is 0.6860, indicating that the two dimensions are independent. SRCC between audio quality and A/V overall quality is 0.7876, and between content consistency and overall quality is 0.7926, suggesting that overall quality is influenced by both audio quality and content consistency.

The distribution of MOSs for each dimension is shown in Fig. 5. MOSs are evenly distributed between 20 and 80 for all three dimensions. Additionally, as illustrated in Fig. 6, we compare the average MOSs of different VTA methods across various audio sources in videos. The audio generated by SVA consistently demonstrates superior quality across all source types. Most VTA methods show better audio quality in the instrument category. In the A/V content consistency dimension, most VTA methods struggle with the fantasy, cooking, people, and scenery categories, where it is challenging to generate audio that aligns with the content of these four types. In the overall A/V quality dimension, VTA methods underperform in the people and cooking categories.

The range of MOSs is [0, 100]. We categorize the video content into 11 main audio sources and then calculate the standard deviation of the overall quality scores among 15 subjects. We compute the average standard deviation for each category, as shown in Tab. 7. The "Fantasy" category shows the highest standard deviation, as it represents unreal scenarios, leading to more diverse interpretations among subjects.

## C. More Details of Experiments

### C.1. Details of Datasets

We conduct experiments on our proposed AGAVQA-3k dataset, as well as on the TTA and TTM datasets. The TTA dataset (Deshmukh et al., 2024) generates 500 audio samples from 100 prompts using 5 text-to-audio generation methods. Subjects were then invited to rate the quality of the audio and its relevance to the provided description. Similarly, the TTM dataset (Deshmukh et al., 2024) generates 500 music samples from 100 prompts using 5 text-to-music generation methods. Subjects were asked to rate the quality of the music and its relevance to the provided description.

The TTM dataset does not include an evaluation of music's aesthetic quality. Since music aesthetic quality assessment is heavily influenced by personal preferences and the subject's taste in different music styles, it is difficult to achieve an objective evaluation. Therefore, in the AGAVQA-MOS, TTA, and TTM datasets, the audio quality dimension primarily focuses on the quality and realism of the audio.

### C.2. Details of Loss Function

We first pre-train AGAV-Rater using text-defined levels. The model predicts a fixed-length text output of 1 token, representing either "Excellent" or "Bad". During pre-training, the cross-entropy loss is used as the objective function, which can be

*Table 8.* Instructions for Audio-Visual LMMs to Obtain Quality Levels for AGAV, TTA, and TTM.

| Input Type | Dimension | Instruction |
|---|---|---|
| AGAV | Audio Quality | \<video>\<audio>Can you evaluate the audio quality in terms of quality and realism? Response with excellent, good, fair, bad, or poor. |
| | A/V Content Consistency | \<video>\<audio>Can you evaluate the audio and video content consistency? Response with excellent, good, fair, bad, or poor. |
| | Overall A/V Quality | \<video>\<audio>Can you evaluate the overall audio-visual quality in terms of audio quality, audio and video content consistency? Response with excellent, good, fair, bad, or poor. |
| TTA | Audio Quality | \<audio>Can you evaluate the audio quality in terms of quality and realism? Response with excellent, good, fair, bad, or poor. |
| | Audio-Text Consistency | \<audio>The text is \<text>Can you evaluate the audio-text content consistency of the audio and text? Response with excellent, good, fair, bad, or poor. |
| TTM | Audio Quality | \<music>Can you evaluate the music quality in terms of quality and realism? Response with excellent, good, fair, bad, or poor. |
| | Music-Text Consistency | \<music>The text is \<text>Can you evaluate the music-text content consistency of the music and text? Response with excellent, good, fair, bad, or poor. |

defined as:

$$L_{CE} = -\frac{1}{N}\sum_{n=1}^{N}\sum_{c=1}^{C} y_{n,c} log(\hat{y}_{n,c}), \tag{2}$$

where $N$ is the number of AGAVs in the batch, $C$ is the vocabulary size, $y_{n,c}$ is the ground-truth label for the $n$-th AGAV at vocabulary position $c$. $\hat{y}_{n,c}$ is the predicted probability output by the model at position $c$ for the $n$-th AGAV. For fine-tuning, we train AGAV-Rater using numerical scores and employ the Pearson Linear Correlation Coefficient (PLCC) loss as the objective. The PLCC loss is defined as:

$$L = (1 - \frac{\langle \hat{s} - mean(\hat{s}), s - mean(s) \rangle}{\|\hat{s} - mean(\hat{s})\|_2 \|s - mean(s)\|_2})/2, \tag{3}$$

where $s$ and $\hat{s}$ are the vectors of MOSs and predicted scores of AGAVs in a batch respectively, $\langle \cdot \rangle$ represents the inner product of two vectors, $\|\cdot\|$ denotes the norm operator for a vector, and $mean$ is the average operator for a vector.

### C.3. Detailed Information of Compared Methods

For the multi-dimensional scoring task, we selected 16 compared methods, covering four types: audio-visual LMMs, AQA, AVQA, and audio-video (text) alignment.

**Audio-Visual LMMs.** We chose 4 latest open-source audio-visual LMM models, including PandaGPT (Su et al., 2023), NextGPT (Wu et al., 2024), VITA-1.0 (Fu et al., 2024), and VideoLLaMA2 (Cheng et al., 2024). We rely on official repositories and use default weights to initialize the models. Then, we input AGAVs, TTAs, and TTM into the models, sequentially asking the quality of each dimension and computing the final predicted scores by Eq. 1. The format of instruction for each dimension is shown in Tab.8.

**AQA.** We selected 4 popular AQA methods, including MOSNet (Lo et al., 2019), STOI-Net (Zezario et al., 2020), NISQA (Mittag et al., 2021), and PAM (Deshmukh et al., 2024). PAM is based on a large language model and does not require training, allowing for direct testing. For all other models, we initialize them with default weights and train them on the AGAVQA-MOS, TTA, and TTM datasets. Since these AQA methods predict a single quality score based only on audio information, we modified the models to output multi-dimensional scores. Specifically, for the AGAVQA-MOS subset, we adjust the final fully connected layer output dimension from 1 to 3, and for the TTA and TTM datasets, we change the output dimension from 1 to 2, allowing the models to predict scores across multiple dimensions.

**AVQA.** We selected 3 latest deep learning-based AVQA methods: DNN-RNT (Cao et al., 2023a), DNN-SND (Cao et al., 2023a), and GeneralAVQA (Cao et al., 2023b). DNN-RNT and DNN-SND are used to predict quality scores for compressed distorted A/V content, while GeneralAVQA is designed for predicting quality scores of real-world distorted audio-visual content, which includes issues such as jitter, overexposure, and motion blur caused during user capturing. All 3 AVQA methods extract features separately from both video and audio, then fuse these features and output a one-dimensional quality score via a fully connected layer. We modified the final fully connected layer's output dimension to 3, enabling the three AVQA methods to be trained on the three-dimensional MOS scores in the AGAVQA-MOS subset.

*Table 9.* Inference latency and throughput of the **AGAV-Rater** on videos on RTX4090. As videos have variable lengths, we set the batch size as 1 for them to avoid the padding cost.

| Video Length (*sec*) | 3 | 5 | 7 | 9 | 11 |
|---|---|---|---|---|---|
| Latency (*ms*) | 157 | 204 | 220 | 256 | 332 |
| Throughput (*video/sec*) | 6.36 | 4.91 | 4.55 | 3.90 | 3.01 |

*Table 10.* 230 AIGC videos content distribution and AGAV-Rater filtering accuracy for each category. We collected these AIGC videos and then used ElevenLabs to generate AGAVs, which were subsequently filtered by AGAV-Rater to select high-quality AGAVs in Section 5.6.

| Metric | Animal | Water | People | Vehicle | Object | Scenery | Sea | Fantasy | Fire | Instrument | Cooking | All |
|---|---|---|---|---|---|---|---|---|---|---|---|---|
| Video Number | 41 | 15 | 20 | 22 | 31 | 19 | 20 | 11 | 15 | 24 | 12 | 230 |
| AGAV-Rater filtering accuracy | 0.78 | 0.87 | 0.85 | 0.77 | 0.81 | 0.79 | 0.80 | 0.82 | 0.80 | 0.83 | 0.75 | 0.80 |

**Audio-Video Alignment.** AVID-CMA (Morgado et al., 2021) is a self-supervised learning approach that learns audio-visual representations from video and audio, achieving highly competitive performance when fine-tuned on action recognition tasks. VAST (Chen et al., 2023) is an omni-modality video-text foundational model capable of processing vision, audio, and subtitles. VALOR (Liu et al., 2024a) is a Vision-Audio-Language Omni-peRception pretraining model that projects vision, language, and audio into a shared common space, enabling alignment across vision-language, audio-language, and audiovisual-language domains. All three models align video and audio at the semantic level, so we fine-tune them on the AGAVQA-MOS subset. We use their encoders to extract audio and video features, then concatenate them and apply a fully connected layer to regress the three-dimensional scores.

**Audio(Music)-Text Alignment.** CLAP (Elizalde et al., 2023) utilizes two encoders and contrastive learning to map audio and text descriptions into a shared multimodal space. TTM-Retrieval (Doh et al., 2023) learns a text-music representation for universal text-to-music retrieval. CLAP, TTM-Retrieval, VAST, and VALOR are all capable of aligning audio and text features. We use their encoders to extract audio and text features, and then apply a fully connected layer to regress and predict audio quality and content consistency scores. We load their pre-trained weights and fine-tune them separately on the TTA and TTM datasets.

## C.4. Details of Audio Preprocessing

AGAV-Rater uses default parameters from VideoLLaMA2 for audio preprocessing. Assuming $T$ video frames are extracted, the steps are:

1. Divide the audio into $T$ segments.

2. Concatenate all segments, then crop or zero-pad to a fixed length.

3. Transform into fbank spectrograms with 128 frequency bins.

4. Use BEATs and an MLP block to extract features from the spectrograms.

5. Concatenate audio, video, and text features, and input them into the LLM.

## C.5. More Details of Enhancing ElevenLabs Results via AGAV-Rater

In Section 5.6, we collected 230 silent AIGC videos and used ElevenLabs to generate audios, aiming to verify that AGAV-Rater can assist ElevenLabs in selecting high-quality audio for users. We conducted a statistical analysis of the content distribution of these 230 AIGC videos, with the results presented in Tab. 10. The analysis shows that the video content is quite diverse, enabling a comprehensive evaluation of AGAV-Rater's performance across different types of video content. We further present the accuracy of AGAV-Rater in identifying higher-quality AGAVs across these categories. As shown in the Tab. 10, AGAV-Rater achieves over 75% accuracy in each category.

## C.6. Cost Analysis of AGAV-Rater

In Tab. 9, we report the inference latency of AGAV-Rater on AGAVs. On a single RTX 4090 GPU, the model can predict scores for 6.36 videos of 3 seconds or 3.01 videos of 12 seconds per second. With a performance 20× faster than real-time, the low latency of AGAV-Rater paves the way for broader real-world applications of LMM-based A/V scoring.

