# OpenReview forum: "AGAV-Rater: Adapting Large Multimodal Model for AI-Generated Audio-Visual Quality Assessment"
_ICML.cc/2025/Conference — ICML 2025 poster_

### Official Review · Reviewer_h1V5 · 2025-03-09

**Overall Recommendation:** 5

**Summary:**

This paper studies the LMM to assess the quality of AI-generated audio-visual content, evaluating AGAVs from three dimensions: audio perceptual quality, A/V content consistency, and overall A/V quality. The authors introduce a novel AI-generated audio-visual quality assessment dataset, AGAVQA, and propose an LMM-based AGAV quality assessment method, AGAV-Rater. The AGAV-Rater demonstrates SOTA performance in multi-dimensional scoring tasks. Compared to 11 audio-visual LMMs (4 open-source LMMs and 7 closed-source LMMs), AGAV-Rater can better select the optimal AGAV.

**Claims And Evidence:**

The claims made in the paper are generally well-supported by clear and convincing evidence. The paper introduces AGAVQA, a large-scale audio-visual quality assessment dataset, and AGAV-Rater, an LMM-based model for evaluating AIGC audio-visual content. The evidence provided includes:
1.	The detailed descriptions of AGAVQA dataset construction.
2.	The performance comparisons of AGAV-Rater across multiple datasets (AGAVQA-MOS, TTA, TTM).
3.	The ablation studies of AGAV-Rater, including pretraining, scoring methods, and multi-dimensional instructions.
4.	The subjective experiments confirm the effectiveness of AGAV-Rater in improving user experience.

**Essential References Not Discussed:**

no

**Experimental Designs Or Analyses:**

The experimental design in the paper appears well-structured and methodologically sound, including the following aspects:
1.Evaluation on multi-dimensional scoring tasks: The study evaluates AGAV-Rater using three dimensions: audio quality, audio-visual content consistency, and overall audio-visual quality.
2. Evaluation on optimal AGAV selection tasks: The performance is measured by answer accuracy in selecting the optimal AGAV.
3. Ablation study on pretraining, scoring method, and instruction design.
4. The subjective experiments confirm that AGAV-Rater enhances the user experience of AGAVs.

**Methods And Evaluation Criteria:**

The proposed methods and evaluation criteria in this paper are well-aligned with the problem of AI-generated audio-visual quality assessment. The proposed model AGAV-Rater outperforms prior approaches in terms of both correlation with human scores and optimal AGAV selection accuracy.

**Other Comments Or Suggestions:**

My suggestions are already included in the weaknesses discussed earlier.

**Other Strengths And Weaknesses:**

The paper’s strengths are as follows:
S1) In terms of originality. The paper establishes an AI-generated audio-visual quality assessment dataset and, based on this, proposes the AGAV-Rater to evaluate the quality of AI-generated audio-visual content. This is a novel contribution to the field.
S2) In terms of rationality. The paper thoroughly validates AGAV-Rater’s performance through extensive experiments, including multi-dimensional scoring, optimal AGAV selection, and enhancing the user experience of Elevenlabs.
S3) In terms of importance. This work fills a gap in AI-generated audio-visual quality assessment, contributing to the advancement of AIGC quality assessment and VTA methods.
S4) In terms of structure. The paper is well-written, well-organized, and clearly structured.

The paper’s weaknesses are as follows:
W1) Lack of detailed introduction of evaluation metrics: The evaluation metrics utilized in the experiments, such as SRCC, PLCC, KRCC, and RMSE, are not introduced in the paper. Providing a brief explanation of these metrics would enhance clarity, especially for readers unfamiliar with them.
W2) Insufficient description of the datasets: The use of datasets is crucial in the experiments, but I do not see a detailed introduction to the TTA and TTM datasets.
W3) Additional experimental data would aid understanding: In Tab. 3, including the accuracy of random selection as a baseline would provide valuable context for understanding the model's performance.

**Questions For Authors:**

I have some questions about the paper：
Q1) The authors conduct cross-dataset experiments on the AGAVQA-Pair subset by training the AGAV-Rater on the AGAVQA-MOS subset. However, the cross-dataset performance of the SOTA methods in Tab. 1 is not shown. For example, how does VAST perform on AGAVQA-Pair when it is trained on AGAVQA-MOS?
Q2) The author selects optimal AGAVs for 230 silent AIGC videos to enhance the user experience. Could you provide more details of these 230 videos? More diverse videos could better validate the generalizability of AGAV-Rater.
Q3) How is the AGAV-Rater method tested on AGAVQA-Pair? Is it evaluated using a multi-input comparison approach or a single-input scoring approach?
Q4) In Equation 1, why do the authors utilize excellent, good, fair, poor, and bad as the standard text quality levels?

**Relation To Broader Scientific Literature:**

The paper makes several key contributions that are well-aligned with broader trends in audio-visual quality assessment and large multimodal models, including:
1.	AGAVQA is the first dataset specifically designed for AI-generated audio-visual quality assessment, distinguishing it from traditional AVQA datasets focused on compression and transmission artifacts.
2.	AGAV-Rater integrates LMMs, improving semantic-level quality assessment, an area where traditional AVQA models are weak.
3.	AGAV-Rater is the first LMM-based model specifically designed for AIGC AVQA, bridging the gap between general LMMs and perceptual quality assessment.

**Theoretical Claims:**

The paper relies on empirical validation through extensive experiments on the AGAVQA dataset, demonstrating performance improvements over baseline methods. The claims regarding the effectiveness of AGAV-Rater are supported by experimental results rather than theoretical derivations.

---

> ### Author Rebuttal · Authors · 2025-04-01
>
> We would like to thank the reviewer for the constructive and valuable feedback. We have addressed your concerns point-by-point below.
>
> **1. Definition of evaluation metrics**
>
> SRCC and KRCC measure the prediction monotonicity, while PLCC and RMSE measure the prediction accuracy. Better AGAVQA methods should have larger SRCC, KRCC, PLCC values, and smaller RMSE values. SRCC can be formulated as:
>
> $SRCC = 1-\frac{6\sum^N_{n=1}(v_n-p_n)^2}{N(N^2-1)}$,
>
> where $v_n$ and $p_n$ denote the rank of the MOSs and predicted score, respectively. For a pair of ranks $(v_i,p_i)$ and $(v_j,p_j)$, the pair is concordant if $(v_i-p_i)(v_j-p_j)>0$, and discordant if $<0$. KRCC is defined as:
>
> $KRCC=\frac{C-D}{N(N-1)/2}$,
>
> where $N$ is the number of AGAVs, $C$ and $D$ denote the number of concordant and discordant pairs, respectively. PLCC and RMSE are calculated as:
>
> $PLCC = \left(\sum^N_{n=1}(y_n-\overline{y})(\widehat{y}_n-\overline{\widehat{y}})\right)/$
>
> $\left(\sqrt{\sum^N_{n=1}(y_n-\overline{y})^2\sum^N_{n=1}(\widehat{y}_n-\overline{\widehat{y}})^2}\right)$,
>
> $RMSE = \sqrt{\frac{1}{N} \sum_{n=1}^{N} (y_n - \hat{y}_n)^2}$,
>
> where $y$ and $\widehat{y}$ denote the MOSs and predicted scores, respectively. $\overline{y}$ and $\overline{\widehat{y}}$ are the mean of the MOSs and predicted scores, respectively.
>
> **2. Introduction to TTA and TTM datasets**
>
> TTA and TTM were proposed by [5]. The TTA dataset generates 500 audio samples from 100 prompts using 5 text-to-audio generation methods. Subjects were then invited to rate the quality of the audio and its relevance to the provided description. Similarly, the TTM dataset generates 500 music samples from 100 prompts using 5 text-to-music generation methods. Subjects were asked to rate the quality of the music and its relevance to the provided description.
>
> **3. More experiments**
>
> On the AGAVQA-Pair subset, AGAV-Rater is tested using a single-input scoring approach. We present the accuracy of fine-tuned audio-video alignment methods on the AGAVQA-Pair subset and the accuracy of random selection:
>
> Method |SonicVisionLM | Frieren | V2AMapper | TIVA | V2A-SceneDetector | STAV2A | SSV2A | ReWaS | All
> :-|:-:|:-:|:-:|:-:|:-:|:-:|:-:|:-:|:-:
> Random | 0.33 | 0.20 | 0.41 | 0.20 | 0.50 | 0.25 | 0.20 | 0.25 | 0.32
> AVIDCMA |0.29 | 0.58 | 0.61 | 0.50 | **0.71** | 0.50 | 0.40 | 0.44 | 0.52
> VALOR | **1.00** | 0.75 | 0.72 | 0.70 | **0.71** | **0.70** | 0.40 | 0.44 | 0.55
> VAST | 0.86 | 0.83 | 0.78 | **0.80** | 0.43 | 0.40 | 0.40 | **0.56** | 0.64
> AGAV-Rater | **1.00** | **0.92** | **0.83** | **0.80** | **0.71** | **0.70** | **0.60** | **0.56** | **0.78**
>
> As can be seen, AGAV-Rater achieves the highest accuracy in each category.
>
> **4. Distribution of AIGC videos in section 5.6**
>
> Thank you for your suggestion. We statistically analyze the distribution of the 230 AIGC video contents in Section 5.6 (in the manuscript), and the results are as follows:
>
> Animal| Water | People | Vehicle | Object | Scenery | Sea | Fantasy | Fire | Instrument | Cooking
> :-:|:-:|:-:|:-:|:-:|:-:|:-:|:-:|:-:|:-:|:-:
> 41 | 15 | 20 | 22 | 31 | 19 | 20 | 11 | 15 | 24 | 12
>
> As shown in the table, the video content is quite diverse, allowing for a comprehensive evaluation of AGAV-Rater's performance across different types of video content.
>
> **5. Explanation of standard text quality levels**
>
> Researchers have found that "good" and "poor" are the most frequently predicted tokens by LMM models when addressing quality issues. We then use the standard text rating levels defined by ITU [6]—excellent, good, fair, poor, and bad—to further refine the quality levels corresponding to these tokens.
>
> **References**
>
> [1] Hayes, A. F. and Krippendorff, K. Answering the call for a standard reliability measure for coding data. Commun. Methods Meas., 1(1):77–89, 2007.
>
> [2] Han, J., Gong, K., Zhang, Y., Wang, J., Zhang, K., Lin, D., Qiao, Y., Gao, P., and Yue, X. Onellm: One framework to align all modalities with language. In CVPR, pp. 26584–26595, 2024
>
> [3] Li, Z., Xu, Q., Zhang, D., Song, H., Cai, Y., Qi, Q., Zhou, R., Pan, J., Li, Z., Tu, V., et al. Groundinggpt: Language enhanced multi-modal grounding model. In ACL, pp. 6657–6678, 2024.
>
> [4] R. I.-R. BT, Methodology for the subjective assessment of the quality
> of television pictures. ITU, 2002.
>
> [5] Deshmukh, S., Alharthi, D., Elizalde, B., Gamper, H., Ismail, M. A., Singh, R., Raj, B., and Wang, H. Pam:Prompting audio-language models for audio quality assessment. arXiv preprint arXiv:2402.00282, 2024.
>
> [6]Recommendation 500-10: Methodology for the subjective assessment of the quality of television  pictures. ITU-R Rec.BT.500,2000.

---

> > ### Comment · Reviewer_h1V5 · 2025-04-02
> >
> > Thanks to the authors for the detailed response. Overall, this paper is essential for the study of AIGC audio-visual quality, and I am willing to increase the score. I hope the above response can be added to the final version. Additionally, there are a couple of minor issues:
> > 1.Does the score in the TTM dataset consider the beauty of music?
> > 2.Could the authors further provide the performance of AGAV-Rater for each category in the 230 videos?

---

> > > ### Author Response · Authors · 2025-04-03
> > >
> > > We sincerely thank the reviewer for reading our response and raising the score. We will include the above responses in the final manuscript. Below are further answers to the reviewer's questions:
> > >
> > > 1. **Evaluation dimensions in the TTM dataset**
> > >
> > > The TTM dataset does not include an evaluation of music's aesthetic quality. Since music aesthetic quality assessment is heavily influenced by personal preferences and the subject's taste in different music styles, it is difficult to achieve  objective evaluation. Therefore, in the AGAVQA-MOS, TTA, and TTM datasets, the audio quality dimension primarily focuses on the quality and realism of the audio.
> > >
> > > 2. **Performance of AGAV-Rater for each category in Section 5.6**
> > >
> > > We categorized the 230 videos in Section 5.6 into 11 categories. Below, we further present the accuracy of AGAV-Rater in identifying higher-quality AGAVs across these categories:
> > >
> > > Animal| Water | People | Vehicle | Object | Scenery | Sea | Fantasy | Fire | Instrument | Cooking | All
> > > :-:|:-:|:-:|:-:|:-:|:-:|:-:|:-:|:-:|:-:|:-:|:-:
> > > 0.78|0.87|0.85|0.77|0.81|0.79|0.80|0.82|0.80|0.83|0.75|0.80
> > >
> > > As shown in the table, AGAV-Rater achieves over 75% accuracy in each category.

---

### Official Review · Reviewer_AFZv · 2025-03-10

**Overall Recommendation:** 2

**Summary:**

This paper introduces a AI-generated audio-visual (AGAV) quality assessment dataset (AGAVQA) and AGAV-Rater, a large multimodal model (LMM)-based approach for evaluating AGAV. The AGAVQA dataset containing two subsets: AGAVQA-MOS (multi-dimensional score prediction) and AGAVQA-Pair (optimal AGAV selection). AGAV-Rater is trained using a two-stage process—pre-training with automatically labeled text-defined quality levels and fine-tuning with human-annotated numerical scores. The model achieves state-of-the-art performance on AGAVQA, text-to-audio (TTA), and text-to-music (TTM) datasets, surpassing traditional AVQA and AQA methods as well as general-purpose LMMs.

**Claims And Evidence:**

The paper does not provide a detailed correlation analysis among Audio Quality, Content Consistency, and Overall Quality dimensions in the AGAVQA-MOS dataset. Such an analysis is crucial for understanding dependencies between these dimensions and would improve interpretability. Without this, it is unclear whether Overall Quality is primarily influenced by Audio Quality or Content Consistency (or both), limiting insights into how AGAV-Rater makes its predictions.

**Essential References Not Discussed:**

None

**Experimental Designs Or Analyses:**

While the paper compares AGAV-Rater to prior LMMs (e.g., VideoLLaMA2), these models cannot be fine-tuned, making the comparison somewhat unfair. It remains unclear whether AGAV-Rater’s advantage comes from its model architecture or its dataset adaptation.

In addition, the audio quality or audio-video alignment could be related to how the audio is processed, e.g., how to encode the audio, how to mix audio and video. I am wondering whether the proposed methods could fit all different processing methods.

**Methods And Evaluation Criteria:**

Audio-video alignment methods (e.g., VAST) achieve strong results on AGAVQA-MOS, raising questions about whether the dataset primarily measures alignment quality rather than broader quality aspects. If AGAVQA-MOS is dominated by alignment factors, then AGAV-Rater’s superior performance might be due to its ability to model alignment, rather than providing a generalizable AGAV quality metric. A deeper analysis of how AGAV-Rater differs from alignment-based methods (e.g., VAST, VALOR) is needed to clarify whether AGAVQA-MOS is capturing a diverse range of distortions beyond alignment.

**Other Comments Or Suggestions:**

1. Several grammatical errors were found and the whole paper could be improved for clarity and readability. Such as
 'It label AGAVs quality in two ways' -> 'It labels AGAVs' quality in two ways';
'Our core contributions can be summarized as three-fold'->'threefold", and better to use either 'Our core contributions are threefold' or 'Our core contributions can be summarized in three ways'.
'AGAV-Rater demonstrates superior score prediction capabilities on the AGAVQA- MOS, TTA, and TTM datasets, and achieves the highest accuracy in identifying the optimal AGAV on the AGAVQA- Pair subset AGAV-Rater offer users a better audio-visual experience and enhance the quality of VTA method outputs.' -> 'offers' and 'enhances'. Moreover, the original combines multiple ideas without proper separation. It would be good to add a comma before the second AGAV-Rater.

**Other Strengths And Weaknesses:**

Strengths:
This paper addresses the quality assessment of AI-generated audio-visual (AGAV) content by proposing a novel dataset (AGAVQA) and a multimodal model (AGAV-Rater), and experimental results show notable improvements over other related methods across multiple evaluation dimensions.

Weakness:
1. The paper lacks a detailed correlation analysis among Audio Quality, Content Consistency, and Overall Quality dimensions in the AGAVQA-MOS dataset, which limits the interpretability of results and insights into dimension dependencies.
2. There is no adequate analysis of the distinctions from AGAV-Rater and alignment methods, even though audio-video alignment methods (e.g., VAST) demonstrate strong performance. Given the fact of VAST's competitive results, it raises questions about whether the AGAVQA-MOS dataset primarily emphasizes audio-video alignment quality rather than broader quality aspects.
3. The human evaluation process are insufficiently described. Critical information such as annotator backgrounds and training procedures is missing, raising concerns about annotation robustness and representativeness. Specifically, the authors do not discuss the annotation quality clearly as well, such as the consistency of ratings across annotators, individual differences in subjective perception, or how potential biases and variances were controlled or mitigated.
4. AGAVQA-Pair dataset evaluation suffers from a notably limited scale (only 75 question-answer pairs) and simplistic annotation (best-of-pair selection), undermining its effectiveness for reliably assessing model generalization.
5. Fig. 2 step1 indicates a significant data issue: the "Kling" video source appears twice with different sample counts, raising severe concerns regarding dataset accuracy and reliability.
6. The paper needs clearer motivation, reasoning, and a stronger discussion of how it differs from prior work. Overall, it is difficult to read.

**Questions For Authors:**

n/a

**Relation To Broader Scientific Literature:**

The paper presents the first large-scale AGAV quality assessment dataset, comprising 3,382 AGAVs from 16 VTA methods, which is of great interest to both industry and academic research communities.

**Theoretical Claims:**

I don't think there are any theoretical claims since the paper focuses on new dataset and how to evaluate the audio-visual quality via a proposed two-stage training processes.

---

> ### Author Rebuttal · Authors · 2025-04-01
>
> We would like to thank the reviewer for the constructive and valuable feedback. We have addressed your concerns point-by-point below.
>
> **1. Correlation of 3-dimensional MOSs**
>
> **SRCC between audio quality and content consistency is 0.6860, indicating that the two dimensions are independent**. SRCC between audio quality and overall quality is 0.7876, and between content consistency and overall quality is 0.7926, suggesting that **overall quality is influenced by both audio quality and content consistency**.
>
> **2. Differences between AGAV-Rater and audio-video alignment methods**
>
> The key difference between AGAV-Rater and VAST is that **AGAV-Rater utilizes the semantic understanding ability of LLM**, improving performance. Although audio-video alignment methods align features semantically, we fine-tuned them on the AGAVQA-MOS subset, mapping audio and video features to quality dimensions. After fine-tuning, these features contain both alignment information and quality information. Therefore, the suboptimal performance of alignment methods does not imply that AGAVQA-MOS emphasizes alignment.
>
> **3. More experiments**
>
> We conduct ablation studies to compare AGAV-Rater with fine-tuned LMMs, with a detailed analysis in Section 3 of the response to Reviewer R3k7. We also test on the AVQA dataset, SJTU-UAV, focusing on real-world user capture distortions.
>
> Method | SRCC | PLCC
> :-|-:|:-
> DNN-RNT (TIP 2023) | 0.7125 | 0.7253
> GeneralAVQA (TIP 2023)  | 0.7753 | 0.7827
> AGAV-Rater | 0.7955 | 0.8052
>
> AGAV-Rater achieves the best performance on SJTU-UAV, proving **its superiority comes from the model framework and training, not from dataset adaptability**.
>
> **4. Details of audio processing**
>
> AGAV-Rater uses default parameters from VideoLLaMA2 for audio preprocessing. Assuming $T$ video frames are extracted, the steps are:
> 1. Divide audio into $T$ segments.
> 2. Concatenate all segments, then crop or zero-pad to a fixed length.
> 3. Transform into fbank spectrograms with 128 frequency bins.
> 4. Use BEATs and an MLP block to extract features from the spectrograms.
> 6. Concatenate audio, video, and text features, and input them into the LLM.
>
>
> **5. Details of human evaluation**
>
> We invited subjects familiar with AVQA and AGAV for on-site training. We provided detailed explanations of the scoring criteria for each dimension and additional AGAV samples for practice. Experts then reviewed the annotations and selected 15 subjects. To prevent fatigue, each subject rated a maximum of 60 samples per day, completing the task in about two months.
>
> We used the **ITU-recommended MOS processing method** [4], and no subjects were identified as outliers. **Krippendorff's α** [1] for audio quality, content consistency, and overall quality are 0.6814, 0.7343, and 0.7143, respectively, indicating appropriate variations among subjects. We also randomly divide subjects into two groups and calculate the **SRCC of average scores between the two groups**. After ten repetitions, the average SRCC for audio quality, content consistency, and overall quality are 0.8043, 0.8318, and 0.8297, validating rating consistency.
>
> **6. Supplement to the AGAVQA-Pair subset**
>
> Due to the lack of public AGAVQA datasets, the AGAVQA-Pair subset was collected from 8 VTA webpages released in the past year. **Its significance lies in that, as a third-party platform, it offers a more objective and impartial dataset.**
>
> **Best-of-pair selection is more reliable than scoring tasks**. Subjects show greater consistency and confidence in determining the optimal sample. In practical applications, identifying the optimal AGAV may be enough. **We use 230 AGAV pairs collected in Section 5.6 (in the manuscript) to further validate generalization.** The accuracy of AGAV-Rater, along with fine-tuned VAST, VALOR, and AVID-CMA, is 80%, 74%, 69%, and 68%, respectively.
>
> **7. Modification of Fig.2**
>
> We apologize for the error in Fig. 2. There are 45 Kling video sources in total, with 23 from the Kling official website, and 22 from the video generation benchmark Vbench. This will be corrected in the final manuscript.
>
> **8. Motivation of our paper**
>
> Previous AVQA and AQA work focused on **real-world capture or compression distortions**. Audio-video alignment methods targeted **semantic alignment in real scenarios**. LMM-based quality assessment **primarily centered on visual quality, with a limited focus on audio**. As AIGC video technology advances, more research explores dubbing techniques. The motivation of our paper is that the quality of AGAVs needs to be monitored and controlled. In the AGAVQA-MOS subset, both audio and video content are AI-generated. We aim to use LMMs to evaluate AGAV quality, replacing human subjective scoring to enhance efficiency and enable automation.
>
> **9. Correction of grammar errors**
>
> Thank you for pointing out our grammatical errors. We will correct these mistakes in the final manuscript.
>
> **References**
>
> Please refer to the Response to Reviewer h1V5.

---

### Official Review · Reviewer_R3k7 · 2025-03-11

**Overall Recommendation:** 3

**Summary:**

This work introduces a new quality assessment dataset and network for the AI-Generated Audio-Visual task. The database additionally handles multimodal challenges like A/V content inconsistency, and the quality assessment model leverages LMM to predict multi-dimensional scores.

**Claims And Evidence:**

The claims made in the submission are supported by detailed experiments.

**Essential References Not Discussed:**

From my opinion, there are no missing essential references.

**Experimental Designs Or Analyses:**

The overall experimental design is mostly fine, with only a few details that need to be confirmed. Also, the ablation experiments can include more base models.

**Methods And Evaluation Criteria:**

The proposed methods and evluation criteria make sense for the assessment of audio-visual quality.

**Other Comments Or Suggestions:**

Please refer to the weakenesses.

**Other Strengths And Weaknesses:**

I do not find major issues in this work overall, except for some minor details:

1.Are there any failure cases during the data auto-labeling process? What is the error rate approximately?

2.Please show the variance in human scores during the subjective experiment.

3.The ablation section could be further expanded, such as by including experiments with different base models in the ablation study.

4.More cases could be shown to further demonstrate the effectiveness, such as the sample and the corresponding score rated by the proposed QA model.

**Questions For Authors:**

Please refer to the weakenesses.

**Relation To Broader Scientific Literature:**

This work may provide some insightful implications for the evaluation of audio-visual quality consistency, the application of large multi-modality models in quality assessment methods, and the development of improved video-to-audio methods.

**Theoretical Claims:**

I checked the proofs and formulas in the method section, and there are no issues.

---

> ### Author Rebuttal · Authors · 2025-04-01
>
> We would like to thank the reviewer for the constructive and valuable feedback. We have addressed your concerns point-by-point below.
>
> **1. Details of the auto-labeling process**
>
> We manually verify 500 auto-labeling results. Among them, the accuracy for content consistency related instruction-response pairs is $100$%, while the accuracy for audio quality related instruction-response pairs is $92$%. In content consistency-related instruction-response pairs, when the consistency quality is labeled as "bad", we ensure that **audio (text) and video from different categories are paired to achieve high accuracy**. In audio quality-related instruction-response pairs, for noisy audio types, such as machine sounds or wind noise, the reverse operation has a minimal negative impact on audio quality. We have **utilized category labels to filter out certain audio quality-related instruction-response pairs**, such as hair dryer drying and pumping water, to minimize the error rate.
>
> **2. Anlysis of human scores**
>
> The range of MOSs is $[0,100]$. We categorize the video content into 11 main audio sources and then calculate the standard deviation of the overall quality scores among 15 subjects. We compute the average standard deviation for each category:
>
> Animal| Water | People | Vehicle | Object | Scenery | Sea | Fantasy | Fire | Instrument | Cooking
> |:-:|:-:|:-:|:-:|:-:|:-:|:-:|:-:|:-:|:-:|:-:|
> 12.56 | 12.16 | 12.20 | 11.79 | 12.57 | 12.74 | 12.37 | **13.11** | 12.35 | 11.93 | 12.23
>
> **The "Fantasy" category shows the highest standard deviation**, as it represents unreal scenarios, leading to more diverse interpretations among subjects. Krippendorff's α [1] can be used to measure the quality of the subjects' ratings. We calculate **Krippendorff's α** for audio quality, content consistency, and overall quality, which are 0.6814, 0.7343, and 0.7143, respectively, indicating appropriate variations among subjects. We also randomly divide subjects into two groups and calculate the **SRCC of average scores between the two groups**. After ten repetitions, the average SRCC for audio quality, content consistency, and overall quality are 0.8043, 0.8318, and 0.8297, validating rating consistency.
>
> **3. Ablation study of the base models**
>
> We conduct ablation studies using GroundingGPT [2] and OneLLM [3] as base models on the AGAVQA-MOS subset. For OneLLM and GroundingGPT, we first load the default weights, and then fine-tune them using the official training code on the AGAVQA-MOS subset. To ensure fairness, we also add the quality regression module, directly regressing the LLM's last hidden states to output three-dimensional numerical scores. The results are as follows:
>
> Dimension | Audio | Quality | Content|Consistency| Overall|Quality
> |:-|-:|:-|-:|:-|-:|:-|
> Metric | SRCC | PLCC | SRCC | PLCC | SRCC | PLCC
> GroundingGPT (ACL 2024)|0.4387 | 0.4494 | 0.5067 | 0.4764 | 0.4975 | 0.5297
> OneLLM (CVPR 2024)|0.6578 | 0.6879 | 0.6184 | 0.6297 | 0.6327  | 0.6388
> AGAV-Rater | **0.7909** | **0.8108** | **0.7553** | **0.7645** | **0.7458** | **0.7552**
>
> As shown in the experimental results, AGAV-Rater achieves the best performance. The main reason for this is that VideoLLaMA2 is designed for audio-video content understanding and pre-trained on more diverse audio-video datasets, making it more suitable for our quality assessment task. GroundingGPT focuses more on localization and visual understanding and is not designed or trained to understand continuous audio-video content. Its ability to comprehend video quality may be weaker. OneLLM is a general multimodal model that, while supporting audio and video processing, is not specifically optimized or enhanced for video and audio alignment. Its audio-related dataset only includes audio-text data, and OneLLM is more suited to text-vision or text-audio matching and understanding, rather than specific audio-video content.
>
> **4. More cases displays**
>
> Thank you for your suggestion. We have displayed more cases about the samples and the corresponding scores rated by the AGAV-Rater on the project page (https://agav-rater.github.io).
>
> **References**
>
> Please refer to the Response to Reviewer h1V5.

---

### Official Review · Reviewer_F7tt · 2025-03-14

**Overall Recommendation:** 4

**Summary:**

This paper addresses a challenging and important question for the VTA methods: whether LMMs can be utilized to assess the quality of audio-visual content generated by VTA methods. To tackle this problem, the authors first establish a large-scale AGAV quality assessment dataset, AGAVQA, which includes two subsets:
AGAVQA-MOS: Contains 9,264 MOS scores for 3,088 AGAVs.
AGAVQA-Pair: Contains 75 question-answer pairs for 294 AGAVs.
Then, this work introduces the AGAV-Rater, a LMM-based quality assessment method for AI-generated audio-visual content. AGAV-Rater can provide multi-dimensional scores for AGAVs, TTAs, and TTMs. Extensive experiments validate the performance of the AGAV-Rater in predicting multi-dimensional quality scores for AGAVs, TTA, and TTM, and assisting VTA methods in selecting the optimal AGAV samples.

### After rebuttal ###
I have read the rebuttal, and my concerns have been well solved. Thus, I tend to keep my original score.

**Claims And Evidence:**

Yes, the authors demonstrate the effectiveness of their proposed model AGAV-Rater in predicting multi-dimensional quality scores on three datasets. This provides clear evidence that their proposed model can adapt LMM for AI-generated audio-visual quality assessment. Additionally, the authors further validate their claims through subjective experiments, showing that their model, AGAV-Rater, can effectively assist video-to-audio methods in selecting the optimal AGAV samples.

**Essential References Not Discussed:**

No

**Experimental Designs Or Analyses:**

Yes, the authors validated their proposed model on three datasets: AGAVQA, TTA, and TTM, demonstrating its ability to provide multi-dimensional scores for AIGC audio-visual, audio, and music content. Additionally, the authors conduct cross-dataset validation experiments, comparing the accuracy of their proposed model with closed-source LMMs in the optimal AGAV selection task. These experimental designs are reasonable and effective. Furthermore, the authors invited participants to verify that the AGAV samples selected by their model enhance the viewing experience, providing a more rigorous validation of the model’s performance through user experience.

**Methods And Evaluation Criteria:**

Yes, the AGAV quality assessment dataset, AGAVQA, established by the authors, contributes significantly to the advancement of the AIGC audio-visual quality assessment field. The model proposed by the authors is theoretically sound and can be effectively applied to evaluate the quality of AI-generated audio–visual content, as well as to identify the optimal AIGC audio-visual samples.

**Other Comments Or Suggestions:**

1.Labeling the boxes in Fig. 2 as (a), (b), and (c) would help readers better understand the figure.
2.The paper should provide a detailed definition of the loss function used during the training of AGAV-Rater.

**Other Strengths And Weaknesses:**

Strengths:
This paper is motivated by the development of VTA methods and LMMs. The authors propose a novel issue in current quality assessment.
1.The authors establish a large-scale AGAV quality assessment dataset that includes AGAVs generated by 16 different VTA methods. The dataset is rich and diverse, facilitating the development of VTA methods.
2.The novel LMM-based AGAV quality assessment method, AGAV-Rater, proposed in the paper, enables multi-dimensional scoring for AGAVs, TTA, and TTM. The authors also conduct extensive experiments, demonstrating that the model can be applied to real-world VTA methods to enhance user viewing experiences.
3.The analysis of multi-dimensional instructions in the experiments is quite interesting. It provides insights for future multi-dimensional quality assessments and can be easily implemented to improve the performance of quality assessment methods with multi-dimensional scoring.

Weaknesses:
The main weaknesses of the paper lie in some unclear explanations, which may confuse readers.
1.The paper lacks some details in the human evaluation section, such as a more detailed display of the scoring interface and the instructions given to subjects in Fig. 2.
2.The paper does not provide details on the time required to train the AGAV-Rater. Understanding the computational cost and training efficiency is crucial for practical implementation.
3.The paper lacks an introduction to the comparison methods. How did the authors train the multimodal alignment-based methods on the AGAVQA-MOS subset in Tab. 1 and Tab. 2? These methods are not specifically designed for quality assessment, and their original structure cannot directly output a one-dimensional quality score.
4.The paper lacks a detailed explanation of the 230 silent AIGC videos selected in Section 5.6. Although the authors demonstrate some AGAV samples on the project page, there is no description of the content distribution of these AIGC videos.

**Questions For Authors:**

1.What specific instructions were given to the human subjects during the testing session? Additionally, how long did it take for the subjects to complete the testing phase?
2.In the 50,952 instruction-response pairs designed in the paper, what is the proportion of each scenario (audio-video, audio-text, and music-text)? Can you also analyze why the consistency dimension shows a relatively large improvement in the TTM dataset during the pre-training step in Tab. 4?
3.The description of the category in Tab. 3 is confusing. How did the authors classify the AGAVQA-Pair subset into 8 categories?
4.AGAV-Rater is trained on the AGAVQA-MOS and demonstrated cross-dataset performance on the AGAVQA-Pair. Are the compared methods in Tab. 3 finetuned on the AGAVQA-MOS? Or are the original model parameters directly utilized?

**Relation To Broader Scientific Literature:**

The primary contribution of this paper lies in constructing a large-scale AGAV quality assessment dataset, AGAVQA, which significantly advances the field of AI-generated audio-visual (AIGC) quality assessment. Previous research has predominantly focused on AIGC images and videos, making this work a pivotal step in addressing the gap in quality assessment for AIGC audio-visual content. Furthermore, the authors propose a LMM-based quality assessment method for AI-generated audio-visual content, providing a novel solution for evaluating the quality of audio generated from video or text inputs. This contribution extends the broader scientific literature on AIGC quality evaluation.

**Theoretical Claims:**

Yes, the process of constructing the AGAVQA dataset is rigorous and well-justified. The proposed model, based on the large multimodal model VideoLLaMA2, is theoretically feasible.

---

> ### Author Rebuttal · Authors · 2025-04-01
>
> We would like to thank the reviewer for the constructive and valuable feedback. We have addressed your concerns point-by-point below.
>
> **1. Details of human evaluation**
>
> We invited subjects familiar with AVQA and AGAV for on-site training. We provided detailed explanations of the scoring criteria for each dimension and additional AGAV samples for practice. Experts then reviewed the annotations and selected 15 subjects. To prevent fatigue, each subject rated a maximum of 60 samples per day, completing the task in about two months.
>
> The official testing phase was conducted in a controlled lab with normal indoor lighting, quiet surroundings, and subjects sitting at a comfortable distance of about 60 cm from the screen. The AGAVs were played at their original resolution. **The scoring interface consisted of three continuous quality rating bars and three navigation buttons.** Each rating bar was labeled with a 1-5 Likert scale. Navigation buttons, including "Prev", "Repeat", and "Next", allowed subjects to switch and replay AGAVs freely. In the final manuscript, we will add images of the scoring interface and detailed documentation of the scoring criteria provided to subjects.
>
> **2. Training duration**
>
> We trained AGAV-Rater on two 96GB H20 GPUs, with training epochs set to $5$ on the AGAVQA-MOS subset, taking approximately **$5$ hours**.
>
> **3. Details of audio-video alignment methods**
>
> Original audio-video alignment methods extract audio and video features using their encoders, then align them into a common vector space. We use these encoders with default parameters to extract audio and video features and then concatenate features. **The concatenated features are fed into a fully connected layer with an output dimension of 3** to predict three-dimensional scores.
>
> **4. Distribution of AIGC videos in Section 5.6**
>
> Due to word limitations, the distribution of AIGC videos is described in Section 4 of the response to Reviewer h1V5. We apologize for any inconvenience.
>
> **5. Modification of Fig. 2**
>
> Thank you for your suggestion. In the final manuscript, we will add (a), (b), and (c) to Fig. 2 to help readers quickly understand its content.
>
> **6. Introduction to the loss function**
>
> We use the PLCC loss to optimize the AGAV-Rater:
>
> $L=(1-\frac{\left<\widehat{s}-mean(\widehat{s}), s-mean(s)\right>}{\lVert\widehat{s}-mean(\widehat{s})\rVert_2\lVert s-mean(s)\rVert_2})/2$,
>
> where $s$ and $\widehat{s}$ are the vectors of MOSs and predicted scores of AGAVs in a batch respectively, $\left<\cdot\right>$ represents the inner product of two vectors, $\lVert\cdot\rVert$ denotes the norm operator for a vector, and $mean$ is the average operator for a vector.
>
> **7. Details of instruction-response pairs**
>
> In the 50,952 instruction-response pairs, the audio-video, audio-text, and music-text scenarios contain 25,592, 19,000, and 6,000 pairs, respectively. Under each scenario, half of the pairs  focus on content consistency, and the other half on audio quality.
>
> **8. Analysis of Table 4**
>
> AGAV-Rater uses VideoLLaMA2 as the base model, and the training set used by VideoLLaMA2 mainly focuses on audio, with relatively less on music. In Table. 2 (in the manuscript), it can be seen that its ability to perceive music-text quality is weaker compared to audio-video and audio-text. Although the music-text scenario has the fewest instruction-response pairs, we repeat these pairs twice during pre-training to increase the learning times of AGAV-Rater for the music-text scenario. Therefore, **the music-text instruction-response pairs enhance the music perception ability of AGAV-Rater, improving the performance of consistency dimension on the TTM dataset.**
>
> **9. Supplement to Table 3**
>
> **The AGAVQA-Pair subset was collected from 8 VTA webpages**, dividing it into 8 corresponding categories. For each category, the optimal AGAV is generated by the corresponding VTA method. We chose to collect the AGAVQA-Pair subset from VTA webpages because these AGAVs, sourced from third-party platforms, offer a more objective and impartial dataset. These VTA webpages are all released in the past year, representing the latest technology in VTA methods.
>
> **The compared methods in Table 3 use their original model parameters** without fine-tuning on the AGAVQA-MOS subset. **We further show the accuracy of the audio-video alignment methods which have been fine-tuned on the AGAVQA-MOS subset**:
>
> Method |SonicVisionLM | Frieren | V2AMapper | TIVA | V2A-SceneDetector | STAV2A | SSV2A | ReWaS | All
> :-|:-:|:-:|:-:|:-:|:-:|:-:|:-:|:-:|:-:
> AVIDCMA |0.29 | 0.58 | 0.61 | 0.50 | **0.71** | 0.50 | 0.40 | 0.44 | 0.52
> VALOR | **1.00** | 0.75 | 0.72 | 0.70 | **0.71** | **0.70** | 0.40 | 0.44 | 0.55
> VAST | 0.86 | 0.83 | 0.78 | **0.80** | 0.43 | 0.40 | 0.40 | **0.56** | 0.64
> AGAV-Rater | **1.00** | **0.92** | **0.83** | **0.80** | **0.71** | **0.70** | **0.60** | **0.56** | **0.78**
>
> As can be seen, AGAV-Rater achieves the highest accuracy in each category.

---

> > ### Comment · Reviewer_F7tt · 2025-04-04
> >
> > Thanks for the detailed responses. My concerns have been well solved. Thus, I am inclined  to increase my score. Additionally, it is recommended to add more details of evaluation and experiments in the revision.

---

### Decision · Program_Chairs · 2025-05-01

**Decision:**

Accept (poster)

**Comment:**

The recommendation is based on the reviewers' comments, the area chair's evaluation, and the author-reviewer discussion.

This paper proposes a new dataset and methodology for AI-generated audio-visual quality assessment. All reviewers find the studied setting novel and the results provide new insights. The authors’ rebuttal has successfully addressed the major concerns of most reviewers. However, Reviewer AFZv replied on April 11th to share some persistent concerns on motivation and presentation, which I believe can be addressed in the final version. Overall, I recommend acceptance of this submission. I also expect the authors to include the new results and suggested changes (especially the feedback from Reviewer AFZv) during the rebuttal phase in the final version.